# Phase-locked constructing dynamic supramolecular ionic conductive elastomers with superior toughness, autonomous self-healing and recyclability

Jing Chen[1], Yiyang Gao[1], Lei Shi[2], Wei Yu[1], Zongjie Sun[1], Yifan Zhou[3], Shuang Liu[4], Heng Mao[1], Dongyang Zhang[1], Tongqing Lu[3], Quan Chen[4], Demei Yu[1] & Shujiang Ding[1] ✉

Stretchable ionic conductors are considerable to be the most attractive candidate for next-generation flexible ionotronic devices. Nevertheless, high ionic conductivity, excellent mechanical properties, good self-healing capacity and recyclability are necessary but can be rarely satisfied in one material. Herein, we propose an ionic conductor design, dynamic supramolecular ionic conductive elastomers (DSICE), via phase-locked strategy, wherein locking soft phase polyether backbone conducts lithium-ion (Li⁺) transport and the combination of dynamic disulfide metathesis and stronger supramolecular quadruple hydrogen bonds in the hard domains contributes to the self-healing capacity and mechanical versatility. The dual-phase design performs its own functions and the conflict among ionic conductivity, self-healing capability, and mechanical compatibility can be thus defeated. The well-designed DSICE exhibits high ionic conductivity ($3.77 \times 10^{-3}$ S m⁻¹ at 30 °C), high transparency (92.3%), superior stretchability (2615.17% elongation), strength (27.83 MPa) and toughness (164.36 MJ m⁻³), excellent self-healing capability (~99% at room temperature) and favorable recyclability. This work provides an interesting strategy for designing the advanced ionic conductors and offers promise for flexible ionotronic devices or solid-state batteries.

Inspired biologically, stretchable ionic conductors with an ion-conducting nature and sensory functions have been widely applied into soft ionotronic devices[1–6], such as stretchable touch panels[7,8], actuators and sensors[9,10], ionotronic diodes and transistors[11], triboelectric nanogenerators[12–14] and others. The majority of current ionic conductors, such as hydrogels and ionogels[2,14–20], come in many flavors with diverse capabilities and limitations. In these systems, large amount of liquid provides free ions mobile environment and covalent crosslinked network contributes to the mechanical strength. However, the presence of liquid leads to poor thermal and electrochemical stability and mechanical deleterious effect[21–23], covalent crosslinked network

[1]School of Chemistry, Xi'an Jiaotong University, Xi'an Key Laboratory of Sustainable Energy Materials Chemistry, State Key Laboratory for Mechanical Behavior of Materials, Xi'an 710049, P. R. China. [2]School of Materials, Sun Yat-sen University, Shenzhen 518107, P. R. China. [3]State Key Laboratory for Strength and Vibration of Mechanical Structures, International Center for Applied Mechanics, Department of Engineering Mechanics, Xi'an Jiaotong University, Xi'an 710049, P. R. China. [4]State Key Laboratory of Polymer Physics and Chemistry, Changchun Institute of Applied Chemistry, Chinese Academy of Sciences, Changchun 130022, P. R. China. ✉e-mail: dingsj@mail.xjtu.edu.cn

results in the irreversibility of the polymer structure[24,25], thereby generating the canonical conflict between ionic conductivity, self-healing capability, and mechanical performance, and becoming unfavorable for the flexible and wearable ionotronic devices. Many researchers have committed to breaking the aforementioned trade-off and constructing versatile ionic conductive elastomers[26–29]. Typical strategy is designing novel polymer molecular structures.

Materials properties depend on its molecular structure. High mechanical strength is mainly derived from the frozen covalent crosslinked network, in which the chain segmental motion is restricted. However, high stretchability requires fewer crosslinking sites and more free mobile chain segments. Employment of supramolecular non-covalent chemistries[30–32] or dynamic covalent bonds[33–35] as reversible crosslinks and sacrificial bonds endow the polymer structure with the reversibility feature, providing materials with self-healing capacity and recyclability to extend their service life and improve their reliability and durability[36,37]. Ions transport in the liquid-free polymer systems relies on the polymer polarity and the segmental motions[38,39]. These different properties originating from different molecular mechanism are generally mutually exclusive[40–43]. Therefore, it is a long-standing challenge to achieve the combination of high strength and high toughness with self-healing capacity and recyclability in a given synthetic ionic conductive elastomer. Most of the as-reported liquid-free ionic conductive elastomers with mechanical versatility and self-healability were obtained by introducing supramolecular hydrogen bonding into the designed ionic conductive polymer networks[28,40–46]. Jia et al. recently synthesized a novel liquid-free ionic conductive elastomer (ICE) hosting lithium (Li+) cations and associated anions via lithium bonds and hydrogen bonds, which features high strength and toughness, self-healing behavior, quick self-recovery, 3D-printability, as well as thermal stability and optical transparency[29]. However, the structural characteristic makes this novel ICE unrecyclable and difficult to repair macroscopic damages.

The compatibility among the ionic conductivity, self-healing, and mechanical properties in the polymer electrolytes has been addressed through several polymer engineering strategies. The most eminent strategy is based on the hard-soft dual-phase block copolymer, in which the hard block (polystyrene, PS) contributed to mechanical strength and the soft block (polyethylene oxide, PEO) was responsible for ion transport[47–49]. In addition, nanoscale-phase separation strategy has been proposed to avoid the occurrence of contradictory properties[50]. Guan et al. proposed a phase-separated structure to settle the conflict between mechanical and self-healing ability, in which the polystyrene provided the increasing modulus, and the terminated amide groups were in charge of self-healing mission[51]. Then, Bao et al. introduced the supramolecular design into a polyurethane (PU) network to overcome the conflict between mechanical robustness and ionic conductivity[52]. Moreover, many design strategies including the combination of supramolecular H-bonding interactions and metal ligand bonds, phase separation, and dynamic hard domains were also achieved[53–56].

In this work, inspired by polymer electrolytes for solid-state lithium-ion batteries, and combining with dynamic supramolecular engineering, we design a novel dynamic supramolecular ionic conductive elastomers (DSICE) via phase-locked strategy, wherein locking soft phase polyether backbone conducts lithium-ion (Li+) transport and the synergistic interaction of dynamic disulfide metathesis and stronger supramolecular quadruple hydrogen bonds in the hard phase contributes to the self-healing capacity and mechanical versatility. The well-designed DSICE possesses high ionic conductivity, good optical transparency, superior mechanical robustness and toughness, excellent autonomous self-healing ability and favorable recyclability. With these desirable traits, we have demonstrated its flexible ionotronic

devices for an ionic conductive substrate, an impedance-based and a capacitive touch sensor.

## Results

### Molecular design and preparation of DSICE

Thermoplastic polyurethane (TPU) system with distinct two-phase morphology is well known to possess fine-tuned structures and microphase separation of soft segments and hard segments. In view of this specific structure of TPU combining with Li-ion transport mechanism, the soft phase polyether was employed to associate/dissociate Li+ and the counterparts and transport ions, while the synergistic effect of the dynamic disulfide metathesis (S-S) and stronger supramolecular quadruple hydrogen bonds (H-bonds) in the hard phase domains was used to regulating the self-healing capacity and mechanical properties. In the case of keeping the structure and function of the soft phase fixed and regulating that of the hard phase, we define this as soft phase-locked strategy. Bis(trifluoromethane)sulfonimide lithium salt (LiTFSI) with large anion group, high ionic conductivity, good solubility, and electrochemical and thermal stability was chosen as conductive lithium salt. The soft phase was chosen as polytetramethylene ether glycol (PTMEG, Mn = 2000 g mol$^{-1}$), in which the loosely coordinating O-Li+ interaction and lower activation energy for ion transport can contribute to higher ion conductivity[57,58]. The hard phase featuring dynamic disulfide metathesis (S-S) and strong supramolecular quadruple hydrogen bonding (H-bonds) was chosen as aliphatic bis(2-hydroxyethyl) disulfide (HEDS) and cyclic 2-ureido-4-pyrimidinone (UPy). Multiple dynamic bonds including disulfide metathesis (S-S), strong cooperative crosslinking H-bonds (UPy-UPy) and weak anti-cooperative crosslinking H-bonds (urethane-urethane, urea-urea, or urea-urethane) were introduced into the polymer backbone, spontaneously form a dynamic supramolecular polymer network (Fig. 1a). The disulfide bonds mainly contribute to the self-healing capacity, the strong crosslinking H-bonds is used for mechanical enhancement while weak H-bonds as sacrificial bonds can dissipate strain energy, increase toughness via efficient reversible bonds rupture and recombination.

First, transparent and colorless dynamic supramolecular elastomers (DSE) were successfully synthesized via condensation polymerization. To systematically study the synergistic interaction of dynamic disulfide metathesis (HEDS) and supramolecular crosslinking quadruple H-bonds (UPy) on the DSE properties, a series of DSEs, which was denoted as DSE-0, DSE-1, DSE-2, DSE-3, was synthesized through increasing the content of UPy from 0 to 30% (mol%). The preparation procedure is shown in Supplementary Fig. 1. PTMEG-based prepolymer was first synthesized with two equivalents of hydrogenated 4,4′-methylenediphenyl diisocyanate (HMDI) and a certain amount of dibutyltin dilaurate (DBTDL) as a catalyst, followed by chain extension using bis(2-hydroxyethyl) disulfide (HEDS) and 2-ureido-4-pyrimidone (UPy) in a given ratio. Compared to hexamethylene diisocyanate (HDI) and isophorone diisocyanate (IPDI), HMDI with the alicyclic isocyanates has large steric hindrance preventing crystallization, resulting in completely amorphous transparent colorless materials. Then, a certain amount of LiTFSI was introduced into the optimal DSE polymers for fabricating dynamic supramolecular ionic conductive elastomers (denoted as DSICE).

### Characterization of DSE

Fig. 1a showed the schematic diagram of the as-synthesized DSE macromolecules. [1]H NMR proved the successful synthesis of DSE, as indicated by the characteristic peaks of PTMEG, HMDI, HEDS, and UPy segments in polymeric backbones (Supplementary Fig. 2). Fourier transform infrared (FTIR) spectra and Raman spectra further confirmed the successful preparation of DSE. The N = C = O disappearance of the peaks at 2260 cm$^{-1}$ and the increasing peaks at ~1660 cm$^{-1}$ and

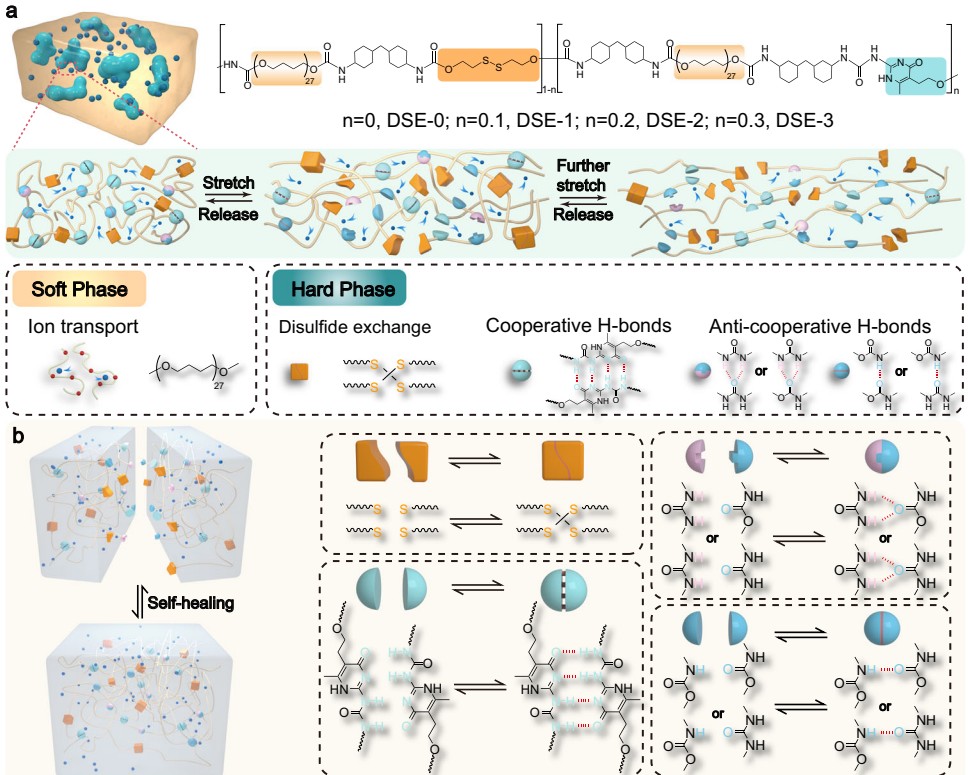

**Fig. 1 | Schematic illustration of the molecular design and mechanism of dynamic supramolecular ionic conductive elastomers (DSICE) with superior stretchability and toughness, autonomous self-healing. a** The chemical structure design and the proposed mechanism for the superior toughness of DSICE upon stretching. **b** The synergistic interaction of multiple dynamic bonds contributes to autonomous self-healing capability.

~1695 cm⁻¹ associated with H-bonded C = O in urea and urethane in the FTIR spectra (Supplementary Fig. 3) indicate that the diisocyanate monomers were completely converted into urethane or urea moieties and the increasing amount of UPy in the DSE polymers. The peaks at 510 cm⁻¹ and 640 cm⁻¹ in the Raman spectra (Supplementary Fig. 4) belong to the υ(S-S) and υ(C-S), respectively, suggesting that disulfide bonds were successfully introduced into the DSE systems. The molecular weight and polymer dispersity index (PDI) of all DSE samples were presented in Supplementary Fig. 5 and Supplementary Tab. 1.

The microstructure of DSE samples was systematically investigated. Fig. 2a presented small-angle X-ray scattering (SAXS) profile plots of DSE specimens. It can be observed that the single broad peak intensity increases with the increasing UPy motifs, indicating more prominent accumulation of the hard segments and the increasing microphase separation domain size on the order of 1–3.7 nm in the materials. Both the electron density contrast between the two phases and the period length of the system increases as the content of UPy motifs in the hard phase increases, as shown in Supplementary Fig. 6. This can demonstrate that the strong hydrogen bonds of UPy provides a stronger driving force for microphase separation[35]. The microphase separated structure of DSE systems was further certified by atomic force microscopy (AFM), as shown in Fig. 2b. The AFM images showed the separation of soft phase (dark areas) and hard phase (bright areas) that is even-distributed and the increasing aggregation of the hard phase with the increasing UPy groups. On account of the microphase separation of the soft and hard domains in the nanometer dimension and the amorphous system, the DSE samples exhibit excellent transparency, as shown in Fig. 2c. The thickness of the DSE films was ~500 μm and the transmittance is higher than 90% in the visible light region. The differential scanning calorimetry (DSC) traces for DSE-0 to DSE-3 was shown in Fig. 2d. The glass transition temperature (Tg) of all DSE samples keeps a very low constant around −76 °C that is the Tg of

PTMEG, suggesting that the lower Tg of DSE polymers derives from the local motions of the soft PTMEG domain and is independent of the UPy content in the DSE backbones. The result fits with "phase-locked" strategy, wherein "locking soft phase" PTMEG realizes ion transport in the DSE polymer systems. This would be discussed later. The thermal performance of DSE was further studied by thermal gravimetric analysis (TGA), which displays that the four DSE samples are thermally stable up to 275 °C in Supplementary Fig. 7.

To gain a deeper insight into the bulk performance of DSE materials, time-temperature superposition (TTS) rheology experiments were carried out. Fig. 2e showed the shear modulus of the DSE materials from 10⁻¹⁰ to 10³ rad s⁻¹. It can be observed that the modulus for the rubber plateau is similar to all DSE. At the crossover frequencies (ωc) between the storage (G′) and loss (G″) modulus is the location where G′(ωc) = G″(ωc), that is, the relaxation time (1/τf) of chain segments, the DSE polymers experience a transition from viscous state to rubber state. At low frequency, G″ is higher than G′, indicative of the viscosity behavior is predominant. As the frequency increases, G′ increases faster than G″, meaning that the elastomer property is ascendant at the ω>ωc region. The increasing amount of UPy in the DSE backbones from 0 to 30% results in a higher rubber plateau and longer chain relaxation time, which corresponds with the increasingly strong UPy H-bonds crosslinking density. Accordingly, the crossover points of G′ and G″ of DSE shifted to higher frequency, indicating that DSE-3 is more rubbery than DSE-0 at higher frequency. These results demonstrate that DSE materials mainly exhibit elastic behavior at room temperature, which is consistent with the lower Tg of DSE. As shown in Supplementary Fig. 8, DMA curves presented temperature dependences of the storage modulus (G′) and the loss factor (tan δ) for the DSE materials. The remarkable drops in G′ relating to the relaxation of soft segments were all distinct and the four strong relaxation peaks appeared in the tan δ curves, which could be assigned to Tg[55]. At

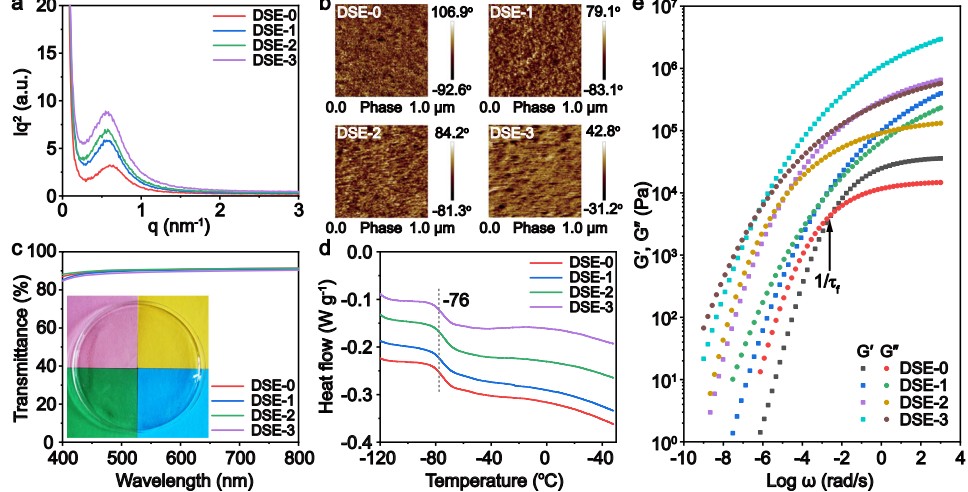

**Fig. 2 | Microphase structures of transparent dynamic supramolecular elastomers with varying amounts of UPy motifs (DSE-0-3). a** SAXS profile plots of DSE-0-3 that demonstrates the presence of microphase separation on the order of 1–3.7 nm. **b** AFM phase images of DSE-0-3. The dark areas represent the soft phase and the bright areas represent the hard phase, indicative of the microphase separation structure of DSE-0-3. **c** Transmittance of DSE-0-3 in the visible range. Insert: photogragh of the as-synthesized DSE, suggesting the high transparency of DSE. **d** DSC traces of DSE-0-3. The constant Tg at around −76 °C is displayed. **e** Time-temperature superposition (TTS) rheology of DSE-0-3. Master curves were built using time-temperature superposition (TTS). Frequency sweeps were performed at 0.1–100 Hz, temperature sweeps were run from 30 °C to 170 °C (30 °C as the reference temperature), and the strain automatically modulated at 5% ± 2% by the instrument to make sure the measured torque at a reasonable value as the sample softened.

temperature above Tg, a second continuous decrease in the storage modulus G′ occurred for DSE samples. This phenomenon was also reported by Kim et al.[59], manifesting that the hard phase domains have a tendency to rearrange at room temperature, which is beneficial to self-healing process.

## Achieving both mechanical and self-healing properties through the synergistic interaction between HEDS and UPy

The mechanical properties of the resulting DSE materials were evaluated by uniaxial tensile tests at room temperature. Fig. 3a showed the tensile stress-strain curves of DSE samples, suggesting that all DSE materials exhibit representative elastic behavior because they do not display yielding phenomena during elongation. Interestingly, both DSE materials exhibited J-shaped stress-strain curves that were strain-dependent mechanical responses. In the unstretched state, the interchain loops lead to the folding of the polymer backbones. At low strains, a line increase of stress occurs, corresponding to the vibration of the molecular links or segments. When keep stretching, the stress increases slowly with increasing strain, corresponding to the extensive soft chain segments, the breakage of dynamic S-S bonds and weak H-bonds. And further stretched, significant strain hardening occurs in the third stage, corresponding to the enhancement of UPy quadruple H-bonds and the unfolding and sliding of the polymer backbones (Fig. 1a). Based on a reasonable structure design, DSE materials possess impressive mechanical properties, such as superior tensile strength, stretchability and toughness, which vary greatly depending on the amount of UPy crosslinks embedded in the polymer backbones. An increase in UPy content contributes to a high improvement in tensile strength and toughness and a slight decrease in stretchability. Without UPy crosslinks, DSE-0 displayed the highest stretchability of 2884.58% and weaker tensile strength of 9.62 MPa. When the UPy content increased to 30 mol%, DSE-3 exhibited optimal mechanical performance with a maximum stress up to 42.60 MPa and a high stretchability of 1630.53%. Toughness, which is related to energy dissipation, has positively correlation with mechanical strength and stretchability. As calculated toughness in Fig. 3b, DSE-3 possessed highest toughness of 259.92 MJ m⁻³. The excellent mechanical properties (strength,

stretchability, and toughness) may be derived from the skillful design of dynamic structures in the hard phase domains.

Cyclic tensile tests of DSE-3 as the optimal mechanical properties were performed to evaluate the self-recoverability of DSE materials at a constant loading/unloading rate of 100 mm min⁻¹. Fig. 3c showed the single cyclic stress-strain curves of DSE-3 at different strains (100%, 300%, 500%, 700%) in the successive tensile process. It can be observed that DSE-3 specimen exhibited a hysteresis loop, a key feature of high toughness. And the larger the tensile strain, the more pronounced the hysteresis loop, which indicated that DSE-3 effectively dissipated strain energy caused by the breaking of dynamic bonds during stretching–retraction cycles at the different strains. Fig. 3d displayed the good deformation recovery of DSE-3 from elongation of 300% and the restoring capability is time-dependent. In addition, six successive loading-unloading cycles tests were performed to further evaluate the self-resilience of DSE-3 at a strain of 300%, as shown in Fig. 3e. The cyclic curves displayed a pronounced hysteresis and a 50% residual strain. The first cycle displayed the larger hysteresis loop and the hysteresis area of the following 2–6 cycles decreased remarkably, which could be attributed to the partially unrecovered dissociated dynamic bonds from the first cycle. Furthermore, after relaxing for 10 min, the loading-unloading cycle curve almost overlapped with the first cycle curve, exhibiting the full recovery of the hysteresis area and the appearance of residual strain. This time-dependent self-recovery property mainly depends on the reversible dissociation/reassociation of relatively weak H-bonds and S-S bonds in the dynamic hard domains. The same is true for the DSE-3 loaded to 500% strain (Supplementary Fig. 9).

To illustrate the crack resistance of DSCE materials, fracture energy was measured to quantitatively assess the crack tolerance via the well-established method for rubber, Rivlin–Thomas pure shear test[60]. To observe the crack propagation, two sets of DSE-3 samples with width of 50 mm and a height of 10 mm were prepared, of which one has 20 mm precut crack (the size of crack was twice the height of the sample), and the other one without crack[61,62]. As shown in Fig. 3f and Supplementary Movie 1, the precut crack was observed to be obvious blunt during stretching. The crack initiated at the front of the notch, propagated in

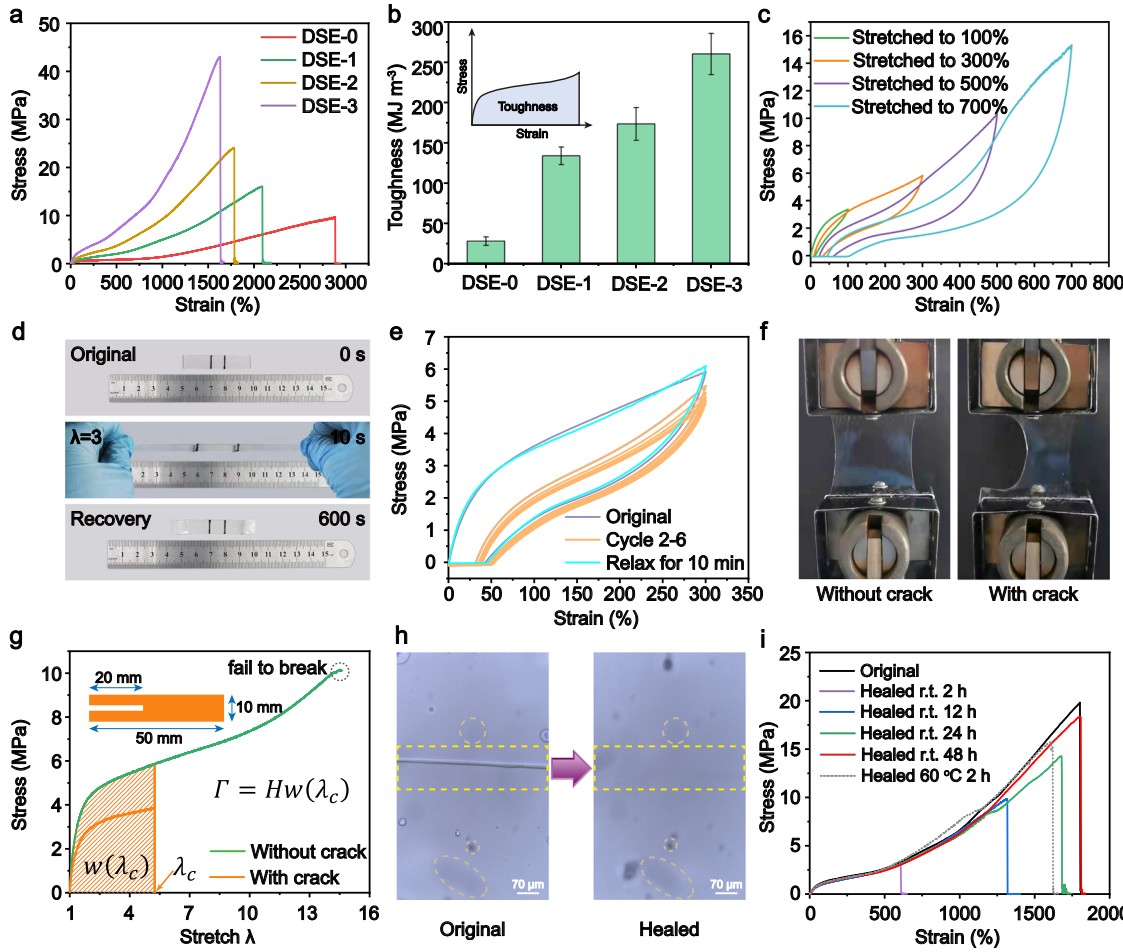

**Fig. 3 | Mechanical properties and self-healing capability of DSE. a** Typical stress-strain curves of DSE-0-3. Maximum strain and stress reach 1630.53% and 42.60 MPa for DSE-3 with a 30% UPy motif, respectively. **b** Toughness of DSE-0-3. It is calculated by the integrate area of the stress-strain curves. Error bars represent the standard deviation calculated by the data sets ($n$ = 3). **c** Successive cyclic tensile behavior of DSE-3 at different strains (100%, 300%, 500%, 700%). The larger the tensile strain, the more pronounced the hysteresis loop, which indicated that DSE can effectively dissipated strain energy (a key feature for high toughness) caused by the breaking of dynamic bonds during stretching–retraction cycles at the different strains. **d** Digital images for DSE-3 film that can restore to its original length after being stretched to 300% strain. **e** Consecutive cyclic tensile curves of DSE-3 at a strain of 300%. After relaxing for 10 min, the cycle curve was fully overlapped with the first cycle, indicating the full self-recoverability of DSE. **f** Photograghs of the intact DSE-3 sample and one with the precut crack stretched to 4 times its original stretching. **g** Stress-strain curves of the intact DSE-3 sample and one with the precut crack (gauge length: 10 mm). The fracture energy ($\Gamma$) is calculated by the formula: $\Gamma = Hw(\lambda_c)$. **h** Optical microscopy images of the scratched and healed DSE-2 film. Healed condition: 12 h at ambient environment. **i** Stress–strain curves of the original and self-healed DSE-2 specimens after different healing time from 2 h to 48 h at r. t. and 60 °C for 2 h. All stretching rate: 100 mm min⁻¹.

the longitudinal direction rather than the transverse direction, and eventually fast fractured. Fig. 3g showed the stress–strain curves of the unnotched and notched samples. The fracture energy[61] of the DSE-3 sample was calculated to be as large as 201.29 kJ m⁻² by the formula of $\Gamma = Hw(\lambda c)$, where $\lambda c$ is the critical stretch of the crack propagation, $w(\lambda c)$ is the elastic strain energy density, which is obtained by integrating the area under the stress-strain curve of the unnotched sample at the $\lambda c$, $H$ is gauge length. The results above demonstrate its exceptional fracture energy, which is caused by the special interior structure of the dynamic hard phase domains.

On account of the synergistic interaction between dynamic covalent disulfide metathesis (HEDS) and reversible supramolecular quadruple H-bonds (UPy)[63,64] in the hard domain, DSE is expected to confer self-healing capability at room temperature. To visualize the excellent self-healing property of DSE materials, Fig. 3h showed that the scratch on the DSE-2 film was observed and finally faded away within 12 h at room temperature. Meanwhile, DSE-2 was chosen to evaluate the full-cut self-healing capability of DSE materials. The original dumbbell DSE-2 was fully cut into two pieces and then put them into contact at ambient condition for different times and at 60 °C for 2 h. Fig. 3i depicted the representative stress-strain curves for the original and recombined films at room temperature for different healing times and at higher temperature 60 °C for 2 h with the tensile rate of 100 mm min⁻¹. The self-healing DSE-2 can be able to reach a 3.07 MPa tensile strength and 604.76% tensile strain, along with 14.76% self-healing efficiency of toughness at room temperature for 2 h. Upon increasing healing time, the ultimate self-healing stress can reach almost 92.76% and the strain mostly overlapped with that of the original sample after 48 h. The self-healing process can be accelerated by raising temperature, while a healing temperature of 60 °C for 2 h brought about the recovery efficiency of 79.80% tensile strength and 89.40% stretching strain. The observed temperature-dependent self-healing performance is attributed to the easier occurrence of dynamic disulfide bonds exchange reaction and the easier recombination of multiple H-bonding interactions in multiphase polymer chains at higher temperature. The self-healing ability of the DSE systems mainly depends on dynamic disulfide bonds and multiple H-bonding interactions between polymer chains (Fig. 1b).

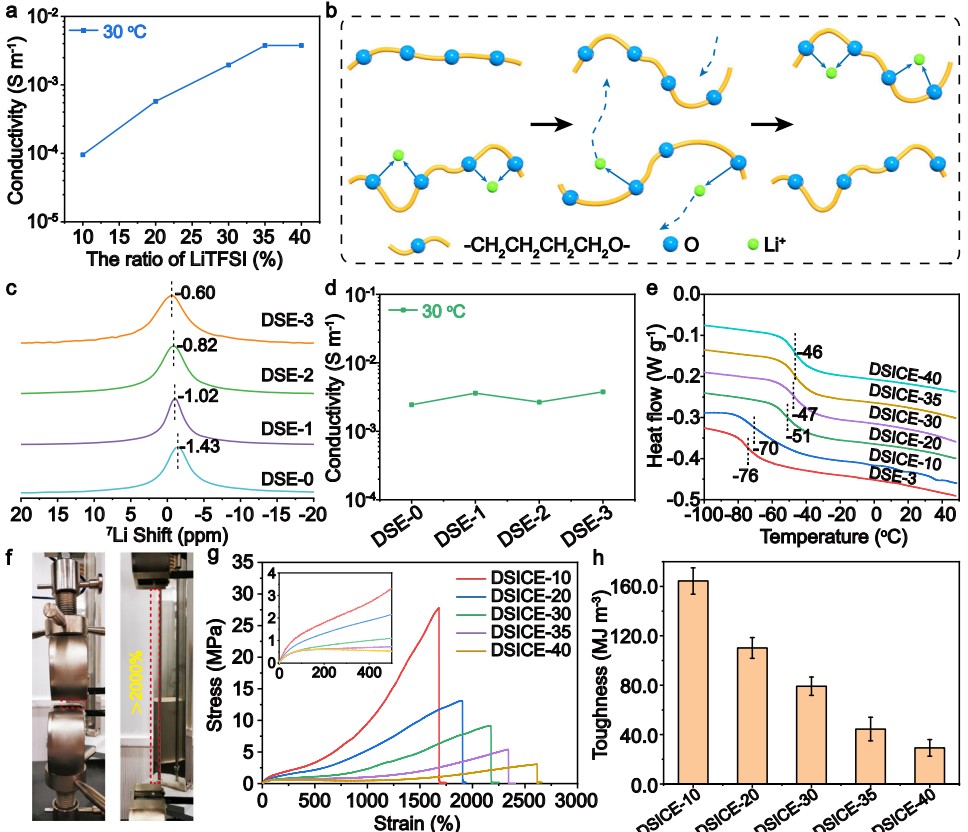

**Fig. 4 | Ion transport mechanism and mechanical properties of DSICE. a** Ionic conductivity of DSICE varying the amount of LiTFSI from 10 wt.% to 40 wt.% at 30 °C. The ionic conductivity of DSICE-35 reaches the maximum value of $3.77 \times 10^{-3}$ S m$^{-1}$. **b** Schematic illustration of Li$^+$ transporting in the PTMEG chains with loose coordination structure. **c** $^7$Li NMR traces of DSICE (DSE-0, DSE-1, DSE-2, DSE-3 with 35 wt.% LiTFSI). **d** Ionic conductivity of DSE-0-3 with the same amount of 35 wt.% LiTFSI at 30 °C. The ionic conductivity of DSE-0-3 with 35 wt.% LiTFSI keeps almost consistent, suggesting that the ionic conductivity of DSICE originate from the soft phase and independent on the hard phase of the polymer backbone, thereby demonstrating phase-locked strategy of DSICE. **e** DSC traces of DSICE-10-40. Tg of DSICE elevates along with the increasing amount of LiTFSI, indicating that the introduction of LiTFSI leads to the restricted movement of polymer chains caused by the coordination of Li$^+$ with the soft PTMEG segments. **f** Digital graph of stretchability of DSICE-30. DSICE can be stretched to >2000% of the original sample. **g** Typical stress–strain curves of DSICE-10-40. Deformation rate: 100 mm min$^{-1}$. Insert: local enlarged view of low strain region of the stress–strain curves. **h** Toughness of DSICE-10-40. Error bars represent the standard deviation calculated by the data sets (n = 3).

## Ion transport mechanism of DSICE

To study the phase-locked strategy, DSICE, denoted as DSICE-10 to DSICE-40, was created by dissolving different amounts of LiTFSI from 10 wt.% to 40 wt.% into the DSE polymers and casting a film (Supplementary Fig. 10). Ion transport properties were investigated through electrochemical impedance spectroscopy (EIS) and DSC traces. Fig. 4a showed the ionic conductivity of DSICE specimens with DSE-3 with 10 wt.% to 40 wt.% LiTFSI. Experimentally, 35 wt.% LiTFSI was chosen to obtain the maximum value, up to $3.77 \times 10^{-3}$ S m$^{-1}$ at 30 °C, calculated by the equation of $\sigma = L/SR$, where $L$ corresponds to the thickness of DSICE samples, $S$ corresponds to the effective overlapping area, and $R$ corresponds to the bulk resistance (Supplementary Fig. 11). The high ionic conductivity mainly owes to the loosely Li$^+$-O coordinating interaction and lower activation energy for ion transport of PTMEG soft polymer chains, as shown in Fig. 4b. DSICE exhibited higher conductivity with the increasing temperature (Supplementary Fig. 12), which can be attributed to more intense movement of polymer chains and ions at higher temperature. Ion transport mechanism was further investigated. Solid-state $^7$Li NMR was employed to study the dissociation environment. Fig. 4c showed the $^7$Li NMR traces of DSICE (DSE-0, DSE-1, DSE-2, DSE-3 with 35 wt.% LiTFSI, respectively). It can be observed that the $^7$Li shifts downfield with increasing amounts of UPy (the increase of highly polar N elements), suggesting that the increase of UPy amounts leads to ion pair dissociation, generating the more dissociated Li$^+$ and the counterparts[65]. Furthermore, ion transport behavior in the DSE polymers was demonstrated in Fig. 4d and Supplementary Fig. 13, which presented the ionic conductivity for DSICE with the same content 35 wt.% LiTFSI in the DSE-0-3 polymers remains relatively constant as the amount of UPy motifs increases. The similarity in the ionic conductivity of the DSE samples suggests that the ion conduction is governed by the soft PTMEG segments and irrelevant to the hard UPy motifs. Fig. 4e depicted the Tg-dependent Li$^+$ transport behavior. The elevated Tg temperature along with the increasing amount of LiTFSI is due to the restricted movement of polymer chains caused by the coordination of Li$^+$ with the soft PTMEG segments. This strongly suggests that Li$^+$ transport environment in the DSE polymers derives from the soft PTMEG segments and the UPy groups do not interfere in ion transport, that is, all the DSE macromolecules transport ions similarly[53,57]. These observations and results demonstrate that the "locking soft phase" realizes ion transport behavior and provides higher ionic conductivity.

## Mechanical properties and reliable performance of DSICE

The mechanical properties of DSICE are an important consideration for the application of flexible ionic devices. Fig. 4f was the photographs of an DSICE-30 test specimen before and after stretching to 2000%. The stress-strain curves of DSICE, inside which was the local enlarged view, were shown in Fig. 4g. It can be observed that all DSICE exhibit

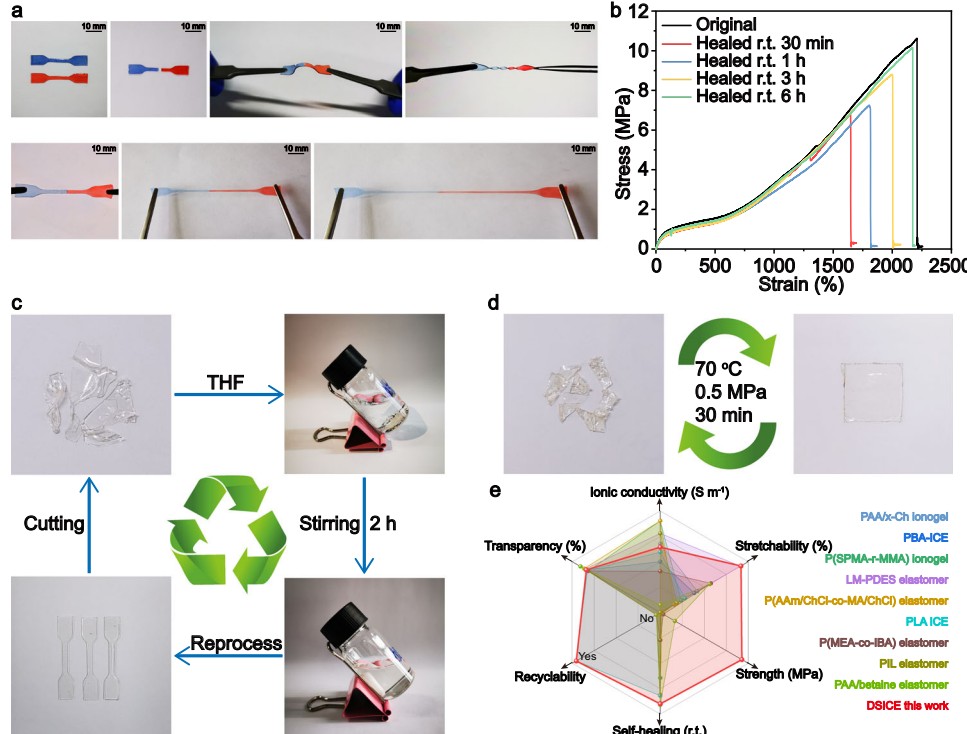

**Fig. 5 | Autonomous self-healing capability and multiple recyclability of DSICE.** **a** Tensile curves of the original and self-healed DSICE-30 sample after different healing time from 30 min to 6h at room temperature. **b** Photograph of DSICE-30 before and after self-healing. The colored dumbbell DSICE-30 was cut into two pieces and then put any two pieces into contact. After self-healed for 10 min, the jointed sample can be bent, twisted and even stretched to 100%, 400%, 800% of the original specimen. **c** Recycling of DSICE. In the direction of arrow: irregular DSICE; irregular DSICE was dissolved in THF and stirred for 2 h to form homogenous solution; Remodeling of DSICE after the solvent completely removed. **d** Recycled DSICE sample via hot-pressing (Condition: 70 °C, 0.5 MPa, 30 min). **e** Comparison of the overall performance between this work and recently reported typical ionic conductor materials.

exceptional strength and stretchability, of which stretching strength can reach up to 27.83 MPa and stretching strain can be more than 2000% of the original length. Addition of LiTFSI salt causes a certain decrease in the mechanical properties of DSICE, which may be due to the large size of the TFSI anions interfering with chain packing and thus preventing aggregation of the hard phase domains[53,66], which was further verified via SAXS characterization (Supplementary Fig. 14). Fig. 4h showed toughness of DSICE, of which DSICE-10 was maximum 164.36 MJ m$^{-3}$ and DSICE-30 reached 79.20 MJ m$^{-3}$. Cyclic tensile tests at different strains of DSICE-20 (100%, 300%, 500%, 700%, 1000%) demonstrated the hysteresis loop became large with the increase of strain, indicating the dissipated strain energy during stretching (Supplementary Fig. 15a). The six successive cyclic tensile curves of DSICE-20 at 300% strain showed an obvious hysteresis loop, but completely recovered to the original curve after a relatively short waiting time (~60 min) (Supplementary Fig. 15b), suggesting DSICE exhibited the time-dependent self-recoverable energy-dissipation network. Rivlin–Thomas pure shear tests were performed to further assess the fracture energy of DSICE materials (Supplementary Fig. 16)[61,62]. The fracture energy of DSE-20 and DSE-30 samples were calculated as 67041.68 J m$^{-2}$ and 23955.94 J m$^{-2}$, respectively, indicating the DSICE materials have favorable crack tolerance and efficient energy dissipation. In this work, the DSICE materials exhibited favorable reliability and long-term performance, as shown in Supplementary Fig. 17. DSICE-20 as a representative can undergo up to 600 cycles at a higher strain of 100% with a certain stress relaxation (Supplementary Fig. 17a), which stems from the break and recombination of the dynamic bonds. It is worth noting that the fast deformation and recovery at the low strain (10−50%) is desirable for flexible devices. Thus, when the strain decreased to 30%, the fresh DSICE-20 and one stored for 3 months cycled up to 375 cycles stably with negligible stress relaxation

(Supplementary Fig. 17b). And DSICE exhibits excellent thermal stability, as shown in Supplementary Fig. 18 and Supplementary Fig. 19.

## Autonomous self-healing capability and recyclable performance of DSICE

The self-healing capability in mechanical performance was investigated. To visualize the excellent self-healing abilities of the DSICE samples, the dumbbell-shaped DSICE-30 film colored blue and red with standard 12 mm × 2 mm rectangular and a thickness of 0.5 mm was cut into two pieces, respectively, then put any two colored pieces into contact and subsequently self-healed quickly at ambient temperature. After self-healing for 5 min, the jointed sample can be bent, twisted, and even stretched to 100%, 400%, 800% of the original specimen, as shown in Fig. 5a and Supplementary Movie 2. The optical microscope image of the self-healed sample was shown in Supplementary Fig. 20, which displayed a seamless combination of two cut-off pieces. The self-healing effect on mechanical performance was quantitatively evaluated by uniaxial tensile experiments. Dumbbell-shaped DSICE-30 specimens were bisected, and then recombined under different conditions. Fig. 5b presented the typical stress-strain curves of DSICE-30 specimens after healing for 30 min to 6h at ambient temperature. When the healing time was over 6 h, the stretch strength and strain of self-healed samples could recover to 10.13 MPa and 2172.64%, respectively, which were almost coincident with original specimens, suggesting that DSICE was endowed with excellent autonomous self-healing capability, which ascribed to the critical contribution of the reversible nature of dynamic S-S bonds, supramolecular H-bond motifs, and extra Li-O dipole interactions[29,62]. From the point of view of the molecular level, the dynamic bonds existing on or near damaged area could promote the chains exchange reaction once contact occurs. Meanwhile, the low Tg of DSICE (<−45 °C) makes

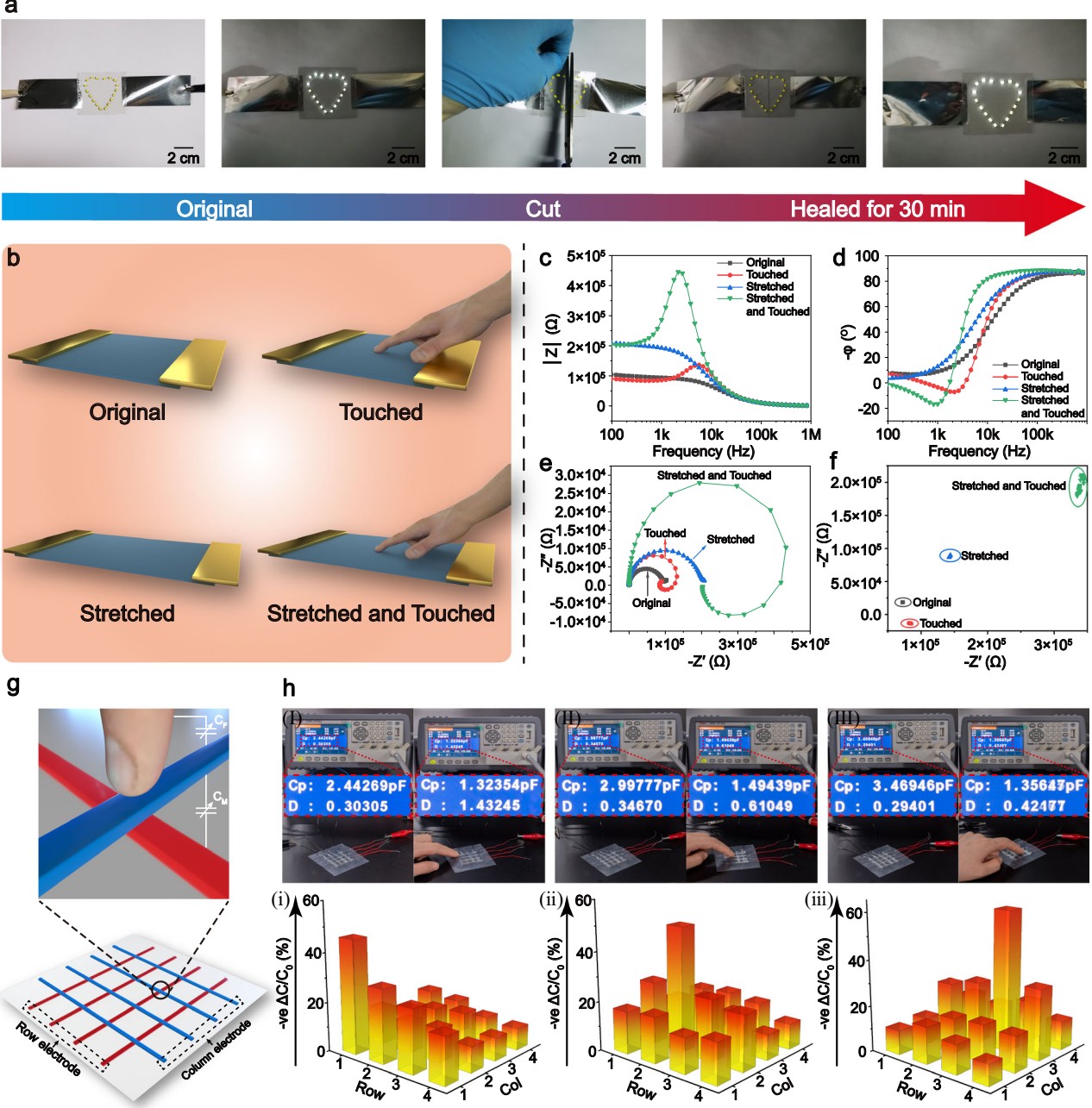

**Fig. 6 | DSICE-based flexible conductive substrate and touch sensor. a** Visually demonstration of DSICE as conductive substrate with excellent self-healing capacity to extend the service life. From left to right: A heart-shaped pattern with 18 LEDs was created; LEDs were lit in an AC 220 V electric field; DSICE-30 sample was cut from the middle of the heart-shaped pattern, LEDs were out; after self-healed for 30 min, LEDs were lit again. **b** Schematic diagram of the DSICE impedance-based touch sensor at different states (original, touched, stretched, stretched and touched). **c** Plots of impedance magnitude (|Z|) versus testing frequency of touch sensor at different states. **d** Negative phase angle (-φ) versus testing frequency of touch sensor at different states. **e** Nyquist plots of impedance spectra of touch sensor at different states. **f** Repeatedly detected data of touch sensor at a single frequency ($f$ = 3 kHz) in impedance complex plane. **g** The work principle and schematic of the DSICE capacitive touch sensor with a 4 × 4 cross-bars array. The finger acts as the third electrode, which capacitively couples to one electrode (blue), as represented by the variable capacitor $C_F$, thus reducing the capacitance between two electrodes $C_M$. **h** The actual test photographs and mapping showing the localized change in capacitance due to a touch by a finger.

them in a high elastic state and enhances the movement of polymer chains at room temperature, which could facilitate self-healing.

The reversible nature of dynamic S-S metathesis and supramolecular H-bonds in the polymer chains contributes to the recyclable performance of DSICE. Therefore, we further studied the recyclable feature of DSICE through recycling in the THF solvent and reprocessing under compression molding conditions. Typically, Fig. 5c showed good recycling of DSICE, that is, DSICE can be fully dissolved into the

THF solvent, recycled by casting the solution and drying, and then reprocessed into the desired specimens. Besides, the small pieces DSICE samples were hot-pressed in a mold applying a force of 0.5 MPa at 70 °C for 30 min, reprocessing into an integrate and coherent film, as shown in Fig. 5d. Interestingly, transparent and smooth films of DSICE were obtained even after reprocessing or recycling for three times. To further reveal the mechanical properties of reprocessed DSICE, tensile tests were performed. As shown in Supplementary

Fig. 21, the reprocessed DSICE-35 showed slightly decreased stretching stress, which is attributed to the insufficient crosslinking of the UPy units after the recycling process. To sum up, DSICE exhibits excellent comprehensive performance, which competes with the highest reported ionic conductors[5,8,9,26–29,46,47], as shown in Fig. 5e and Supplementary Tab. 2.

### DSICE-based flexible conductive substrate and touch sensor

Transparent DSICE with highly competitive properties can be used to fabricate flexible iontronic devices, such as ionic conductive substrate and touch sensor. Based on the high ionic conductivity and transparency, the demo with the LEDs over the flexible DSICE substrate was created, as shown in Fig. 6a. We designed a heart-shaped pattern with 18 chip LEDs on the DSICE-30 substrate. The poles of the chip LEDs were embedded into the DSICE ionic conductor, and the chip LEDs keep a parallel relationship with the ionic conductor. The prepared heart-shaped pattern of 18 chip LEDs formed a closed circuit with the DSICE under an AC 220 V electric field at the fixed frequency of 50 Hz. As expected, the heart-shaped pattern with LEDs can be entirely lit. When DSICE-30 was cut from the middle of the heart-shaped pattern, the LEDs on either side of the heart-shaped pattern could not be lit. Interestingly, the cut-off DSICE-30 was contacted and self-healed for 30 min under ambient condition, on which the heart-shaped pattern with 18 LEDs would be lit up again, and the luminescence intensity did not decay, which demonstrates that DSICE-based conductive substrate with excellent self-healing capacity could extend the service life of devices.

DSICE as ionic conductor has the frequency-dependent feature. Thus, an impedance-based electrical touch sensor was built, the two ends of DSICE ionic conductor were connected to electrodes and incorporated into an AC circuit. Fig. 6b was the schematic diagram of DSICE-based touch sensor at four different stimulus states: original, touched, stretched, stretched, and touched. Actually, human body acts as an ionic conductor. When a finger as part of human body touches the sensor, the human body become part of the circuit, the finger can conduct current and introduce new elements into the circuit, causing a significant change of impedance signal in the AC the circuit characteristics[8]. Hence, we used the impedance spectra and impedance complex plane to recognize different stimulus of the touch sensor. Based on the frequency-dependent feature of DSICE, the impedance spectra can change at different stimulus states over the whole frequency range (1 MHz to 0.1 Hz). As shown in Fig. 6c, d, when the sensor was stretched, the |Z| (impedance value) increased, and the -ϕ (negative phase angle) changed slightly; when touched, the |Z| and -ϕ versus frequency curves showed significant discrepancy from original or stretched states. And the |Z| exhibited a large peak in the frequency range of 100 Hz–1 MHz, the -ϕ changed a lot from positive value (-87°) to negative value (-−20°) in the same frequency range. Obviously, the -ϕ became negative values at low-frequency range (10 kHz–100 Hz), so the touched sensor exhibited inductance characteristics. Fig. 6e presented the differences in the complex plane in the Nyquist plots of the impedance spectra under the four different states. When touched regardless of being stretched or not, the Nyquist plots demonstrate -(-Z″) in specific frequency range. In contrast, the DSICE with touch regardless of being stretched or not appears a negative -ϕ and -Z″ at a certain frequency range. Therefore, we set a single frequency (f = 3 kHz) to detect signals from different stimulus in impedance complex plane, as shown in Fig. 6f. In this way, different stimuli will appear in different region in the two-dimensional impedance complex plane, indicating the real-time responses of the touch sensor to different stimuli. In addition, Supplementary Fig. 22 depicted the impedance changes of DSICE-30 when stretched to different tensile elongation at a fixed

frequency of 1 kHz, as seen in Supplementary Movie 3. And Supplementary Fig. 23 depicted the impedance changes of DSICE-30 at different touched stimuli when stretched at the frequency 1 kHz, as seen in Supplementary Movie 4.

Furthermore, we presented a capacitive touch sensor to realize the position detection of touch sensing. As a proof-of-concept, the 4 × 4 cross-bars sensor array with a 10-mm pitch was demonstrated, which is composed of transparent DSICE-30 electrodes capacitively coupled through a thin silicon film (100 μm), as shown in Fig. 6g. And the capacitive touch sensor works as follows, each array intersection is composed of two electrodes (one is red and the other blue) separated by a dielectric layer. The finger acts as the third electrode, which capacitively couples to one electrode (blue), as represented by the variable capacitor $C_F$, reducing the coupling between two electrodes $C_M$[67]. Therefore, the presence of a finger causes a drop in the capacitance ($C_M$) between the electrodes. In this implementation, the 4 × 4 cross-bars array can be made up to eight lines with four on each axis, generating 16 pixels. A finger touched a fixed pixel, the capacitance of each combination of row and column electrodes, which was connected with the LCR meter, was detected sequentially. The resulting capacitance changes of all pixels were determined to create a mapping. In this case, the maximum capacitance change value in the mapping means the position where a finger touches. Fig. 6h showed the mapping of a finger touching different positions. It can be observed that the capacitance changes at the intersection of Row 1 and Col 1, that is pixel (1,1), Row 2 and Col 2 (2,2), and Row 3 and Col 3 (3,3) are maximum from Fig. 6h (i), (ii), (iii), and the capacitance decreases by 45.9%, 50.1%, 60.9% from Fig. 6h (I) (II) (III), indicating a finger touched pixel (1,1), (2,2), and (3,3), separately. The neighboring pixels exhibited a certain sensitivity due to the fringe field of the finger. In general, whether it is the relatively small minority impedance sensor or the pervasive capacitance sensor, DSICE exhibited the excellent touch sensing feature.

## Discussion

In summary, based on the transport mechanism of lithium-ion in polymer and the dynamic supramolecular engineering, we developed a novel ionic conductor, dynamic supramolecular ionic conductive elastomers, DSICE, by locking soft phase polytetramethylene ether glycol (PTMEG) conducting ion transport and regulating the dynamic metathesis HEDS and supramolecular quadruple UPy H-bonds in the hard phase achieving the excellent mechanical properties and self-healing capability. In addition, the rational molecular design endows DSICE with high optical transparency and favorable recyclability. With these excellent attributes, we demonstrated its applications for a flexible conductive substrate, an impedance-based, and a capacitive touch sensor, which have potential practical value for flexible display and touch fields. Our elaborate designed DSICE provides an exciting avenue to develop advanced ionic conductors with multiple functions and offers a promising material for the emerging flexible and wearable ionotronic devices or solid-state batteries.

## Methods

### Synthesis of dynamic supramolecular elastomers (DSE)

DSE were synthesized via condensation polymerization. PTMEG (1 mmol, RHAWN) in a dried glass vessel was heated and stirred in an oil bath at 120 °C under vacuum (<133 Pa) for 1 h to remove any moisture, and then cooled to 65 °C. HMDI (2.1 mmol, Aladdin, 90%) dissolved in dry DMF (10 mL, $H_2O$, $O_2$ < 50 ppm, obtained from a VSPS-5 solvent purification system) and one drop of dibutyltin dilaurate catalyst (DBTDL, TCI, tech. 95%) was dropwise added into the vessel and stirred for 1.5 h under $N_2$ atmosphere to yield prepolymer. After synthesis of the prepolymer, the final products were obtained via adding different ratios of Bis(2-hydroxyethyl) disulfide (HEDS, 1 mmol, Alfa Aesar, tech. 90%) and 2-ureido-4-pyrimidone (UPy, Synthesized as Supplementary

Fig. 24 and Supplementary Fig. 25) in the same polymerization procedure. HEDS (1 mmol) in appropriate amount of anhydrous DMSO (RHAWN, 99.7%) was added to the prepolymer solution. With fully stirring at 75 °C for another 9 h, a certain methanol (Aladdin, 99.5%) was added and the mixture was further stirred for 30 min to ensure enough consumption of isocyanate groups. Then, the as-prepared solution was poured into a glass plate with subsequently putting in an oven at 60 °C to slowly evaporate the solvent overnight. Finally, the resulting film, named DSE-0, was dried in a vacuum oven at 60 °C for 24 h to obtain a transparent, colorless elastomer. Similarly, DSE-1, DSE-2, and DSE-3 were obtained by adding HEDS (0.9 mmol) and UPy (0.1 mmol), HEDS (0.8 mmol) and UPy (0.2 mmol) and HEDS (0.7 mmol), and UPy (0.3 mmol) in appropriate amount of dry DMSO.

## Preparation of dynamic supramolecular ionic conductive elastomers (DSICE)

DSE polymers were dissolved in appropriate amount of THF ($H_2O$, $O_2 < 50$ ppm, obtained from a VSPS-5 solvent purification system) along with different amounts of vacuum-dried Bis(trifluoromethane) sulfonimide lithium salt (LiTFSI, Aladdin, 99.95%). After fully dissolved, the viscous transparent solution was degassed, cast into a glass mold, and allowed to slowly evaporate at room temperature overnight. Then the resulting film was further dried for 24 h at 60 °C in a vacuum oven. The obtained films were 50–500 μm thickness.

## Materials characterization

ATR-FTIR was performed on a Nicolet 6700 IR spectrometer (Thermo Scientific, USA) with a resolution of 4 cm$^{-1}$ and scanning time of 32. $^{1}$H NMR (400 MHz) spectra were recorded on a Bruker AVANCE III spectrometer. Solid-state $^{7}$Li NMR measurements (600 MHz) were conducted using a JNM-ECZ600R spectrometer (JEOL RESONANCE Inc., Japan). Raman spectra (Renishaw) were recorded on a microscope using a laser excitation wavelength of 532 nm. GPC tests were conducted using a Shimadzu LC-20AD GPC system. THF was used as the mobile phase with flow rate of 1 mL min$^{-1}$. DSC experiments were carried out under $N_2$ atmosphere using a TA Instrument DISCOVER DSC 250 system at a scan rate of 10 °C min$^{-1}$. SAXS measurements on polymer films were performed on a SAXS point 2.0 instrument (Anton Paar, Austria) equipped with a Cu/Mo dual microfocal X-ray source and a two-dimensional hybrid photon counting detector (EIGER R 1M). Transmittance of the films were carried out using UV-vis spectrophotometer (PE Lambda950, China) with the wavelength for testing ranging from 800 to 400 nm. The microphase structure of the samples were measured by tapping-mode AFM (Bruker, INNOVA) using tapping MPP-rotated cantilevers with silicon probes. The thermal stability property was evaluated by thermogravimetric analysis (TGA) with a differential thermal analysis instrument (METTLER TOLEDO TGA/DSC3+) over the temperature range of 25–600 °C in $N_2$ atmosphere with a heating rate of 10 °C min$^{-1}$ (empty $Al_2O_3$ crucible as the reference). The rheological behavior was carried out on a MCR302 (Anton Paar, Graz, Austria) rheometer. Frequency and temperature sweeps were performed with 25 mm parallel plate attachment. Dynamic mechanical analysis (DMA) was performed on a Netzsch DMA242E system with a tension mode.

## Mechanical properties tests

The mechanical performance was mainly evaluated via stress-stain curves. The stress–stain experiments were performed using a TianYuan (TY-8000) Electronic Universal Test Instrument with a 50 N force transducer. The whole samples were cut into a dumbbell shape (a testing size of $12.0 \times 2.0 \times 0.5$ mm$^3$) for tensile tests. All the testing rate was set at 100 mm min$^{-1}$.

For fracture energy tests, pure shear tests were performed with 1 kN load cell. This method required two sets of samples with the same size of the width of 50 mm, the thickness of $0.7 \pm 0.2$ mm, and the gauge height of 10 mm. One set of samples were precut with a crack of 20 mm width, the other set was without crack. The fracture energy value ($\Gamma$) is calculated by the formula:

$$\Gamma = Hw(\lambda c)$$

where $\lambda c$ is the critical stretch of the crack propagation, $w(\lambda c)$ is obtained by integrating the area under the stress–strain curve of the unnotched sample at the $\lambda c$, $H$ is gauge length. The results were repeated for three times on different batches of samples.

## Ion conductivity investigation

Ion conductivity of DSICE was calculated by the following formula:

$$\sigma = \frac{L}{RS}$$

where $L$ (cm) is the thickness of DSICE membrane ($240 \pm 20$ μm and measured by the thickness gauge), $R$ (Ω) is the bulk resistance of DSICE membrane obtained from EIS, and $S$ (cm$^2$) is the contact area of the electrode and DSICE membrane. The DSICE membranes were sandwiched between two stainless steel (S-S) electrodes with a diameter of 16 mm. EIS was measured using a ParStat4000 electrochemical workstation (Princeton, America) over the temperature range from 30 °C to 70 °C, and the frequency range was set from 0.1 Hz to 1 MHz with amplitude of 10 mV.

## Self-healing capability measurements

The visual self-healing results were observed by optical microscopy (OM). The complete fracture self-healing ability was evaluated by stress–stain curves. The dumbbell samples were completely cut in half, then the two pieces of all samples were manually contacted. After self-healing at different conditions, the healed samples were subjected to tensile tests.

## Fabricating of touch sensors

The impendent touch sensor was fabricated by DSICE with a shape of $50.0 \times 25.0 \times 1.0$ mm$^3$. Two aluminum electrodes were then attached at both ends of the DSICE for testing. For the touched measuring, the finger touched the sensor without pressing. The sensor was stretched to one time the original length for the stretched measuring. The capacitive touch sensor with a $4 \times 4$ cross-grid array was fabricated by two electrodes with a 10-mm pitch capacitively coupled through a thin silicon film (100 μm). A finger touched a fixed pixel, each combination of row and column electrodes was scanned sequentially.

## Impedance and capacitance tests

For impendent touch sensors without any pattern design, the impedance tests were performed on an electrochemical workstation (CHI660E), testing $Vrms$ was set at 0.5 V. For capacitive touch sensors, the capacitance change was measured by LCR meter with the voltage of 1 V and the frequency of 20 kHz.

## Data availability

All data generated in this study are provided in the Source Data File. Source data are provided with this paper.

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

## Acknowledgements
We acknowledge Mr. Zijun Ren and Mr. Chang Huang at the Instrument Analysis Center of Xi'an Jiaotong University for assistance with AFM and SAXS. This research was supported in part by the National Natural Science Foundation of China (No. 51973171 [Ding]), and Young Talent Support Plan of Xi'an Jiaotong University [Ding]. Innovation Capability Support Plan of Shaanxi (No. 2022TD-27 [Ding]), China Postdoctoral Science Foundation (2019M663687 [Yu]), Fundamental Research Funds for the Central Universities (xhj032021014-02 [Zhang]).

## Author contributions
The idea and project were conceived by J.C. and S.D.; J.C. designed and conducted the experiments; Y.G. and L.S. contributed to the guidance and implementation of the application section. Y.G., W.Y., H.M., and D.Z. assisted in the mechanical tests. Z.S., D.Y., and S.D. made instructive advice to revise the full text; Y.Z. and T.L. provided assistance in the tests and analysis of fracture energy; S.L. and Q.C. assisted in the TTS fitting and analysis. J.C. and S.D. analyzed all the data and co-wrote the paper. All authors have given approval to the final version of the manuscript.

## Competing interests
The authors declare no competing interests.
