## [Peer Review File · Nature Communications]

Phase-locked constructing dynamic supramolecular ionic conductive elastomers with superior toughness, autonomous self-healing and recyclabilityReviewers' Comments:

Reviewer #1:

Remarks to the Author:

This manuscript reported a novel ionic conductor, dynamic supramolecular ionic conductive elastomers (DSICE) via "phase-locked" strategy and explored the application in flexible iontronic devices. Besides, the performance of DSICE is very competitive compared with as-reported ionic conductors. Based on molecular engineering, the design strategy solves the multiple contradictions of the existing ionic conductors. I think this strategy is innovative and interesting, which is meaningful for the further design of the advanced ionic conductors. Therefore, this manuscript can be considered for publication in Nature Communications after minor revision. The detailed comments are listed below:

1. DSICE exhibits excellent mechanical properties, such as high stretchability, high strength and high toughness. How about its resilience?
2. In the paper, the content of UPy is increased to 30% mol. Whether the increase of UPy will further improve the mechanical strength? It is known that the solubility of UPy is very poor. Will further increasing the content of UPy affect the reaction process?
3. The conductive mechanism of DSICE is the dissociation of LiTFSI by the soft phase PTMEG and the movement of the chain segment to promote the transfer of lithium salt. So, will the ionic conductivity performance be better if the lithium salt is replaced with an ionic liquid? Please explore it.
4. Comparing Figure 2a and Figure 4e, the addition of LiTFSI leads to a certain decrease in the mechanical strength. What is the reason?
5. The author mentioned that DSICE has the potential as a solid-state battery application several times in the article. I recommend that the authors try it in this application.

Reviewer #2:

Remarks to the Author:

In this manuscript, the authors reported the design and construction of a novel ionic conductor, dynamic supramolecular ionic conductive elastomers (DSICE) via the promising "phase-locked" strategy. The strategy skillfully solves the conflicts between ionic conductivity, self-healing capacity and mechanical properties of existing ionic conductors and endows the well-designed DSICE with excellent comprehensive properties, which helps to expand the application scenarios of ionic conductors in stretchable sensors. Overall, the research was well designed and completed at a high level. Therefore, the manuscript would be suitable for publication in Nature Communications after the following questions are well addressed.

1. As we know, pure polymers cannot conduct ions as easy as liquid medium. However, as for the liquid-free ionic conductors in this work, the DSICE exhibited high ionic conductivity. What is the reason behind the good performance? It's better to highlight this aspect because it is very important for potential or even practical applications in various fields including flexible sensors and all solid state batteries.
2. The authors claimed that "Li⁺ dissociation environment in the DSICE materials derive from the soft PTMEG segments and the UPy groups do not interfere in the Li⁺ dissociation and transport" in the Lines 223 and 224.
It is understood that highly polar atoms are conducive to the dissociation of lithium salts, and N and S elements contained in the designed structure may also contribute to the dissociation of lithium salts. Please discuss whether it is possible to happen.

3. The pictures in Figs. 3c and 6a should have scale bars. Besides, there are several mistakes in the main text. For example: please change "combing" into "combining" in line 237; please change "." into "," in line 261; please change "amount" into "amounts" in line 208.

4. The authors claimed that "as the amount of LiTFSI increases, the mechanical strength decreases and the stretch strain increases. This is due to the loose polymer chains stacking caused by the increasing TFSI anions". Are there any evidences to support the hypothesis?

5. The demo of flexible and stretchable touch sensor is very impressive, which takes full advantages of the merits of DSICE including conductivity, flexibility, stretchability and mechanical robustness. However, for the reviewer and potential readers, it's not clear that what's the working principle or mechanism of DSICE-based touch sensor. It is suggested to provide more details on this.

6. The synergistic effect between S-S bond and UPy H-bonds play an important roles in both conductivity and mechanical performance. However, there is a lack of deep discussion about the synergy. How it works or how they work together to boost the performance of the DSICEs.

7. As for the self-healing of DSICE, apart from the explanations of dynamic H-bonding and S-S bond, the reviewer thinks the Li-O dipole interactions are also contributable to the fast self-healing. In fact, it is an extra bonus of the smart design.

Reviewer #3:

Remarks to the Author:

The authors here report on a highly stretchable, self-healable ionically conducting elastomer. Certainly, those types of materials can find a plethora of applications in intronic devices. Hence, there is a rich research background existing on those types of elastomers.

In this paper, the authors show high levels of stretchability in their thermoplastic polyurethane based system. However, as typical for such systems, this behavior is mostly viscoelastic/plastic in nature, with little reversibility. In fact, the authors do not show restoring capabilities of their material when stretched to >300% tensile.

Sadly, this is somewhat typical in this research field, however placing strong emphasis on large but irreversible stretchability holds little promise for practical applications. Instead, a stronger focus on characterizing the "moderate" strain region (here up to 300%, one curve up to 500% in the SI) more exhaustively would be required. As shown in figure 3b, the material exhibits strong hysteresis effects (which the authors acknowledge and favor as this is associated with energy dissipation and "toughness") of the material, but also demonstrates considerable irreversibility of deformation (plastic deformation) of 50% strain. This should be discussed in more detail. 5 stress-strain cycles are not sufficient to study the long-term mechanical behavior of this material and compare it to other systems like hydrogels with ionic liquids.

In general, materials reliability and long-term performance is not assessed here at all.

In addition, the claim on "superior toughness" is not at all substantiated. I assume the authors refer to fracture toughness of the material. However, they simply integrate the stress-strain curve up to rupture of their elastomer (Figure S8). A more proper investigation on crack propagation/crack opening and the associated energies is required. I refer here to the seminal work of Zhigang Suo's group on the topic of fracture toughness in soft materials.

The re-usability part (Figure 5) lacks characterization, and essentially only consists of a couple

photographs. Where are mechanical tests on re-processed material samples? Is it possible to repeat this process? Some data at least should be shown to support the argument.

Finally, the demonstrations of the material as ionic conductor are very rudimentary at best. The demo with the LEDs is in the presented form not understandable. There seems to be some LEDs placed onto the ionic conductor, then a "electric field" is applied. How does this work in detail? Are there structured conductive traces somewhere, or is there just a high AC field applied? Not very exhaustive description of the experiment. Then, the "touch panel" is quite undercharacterised as well. No photographs or proper touch measurements are given. Nor proper mechanical characterization of that panel. Compared to what is out there in literature on ionic touch panels, this demo falls way short. Quite some reworking of the demo section is required to convincingly show the feasibility of this material in practical applications, and its superiority compared to other approaches.

In summary, while the material itself is interesting, and the synthesis seems to be done well, this paper lacks essential information and needs major overhaul. At the current stage, a more specialized chemistry journal would seem a better fit.

Response to Reviewers' Comments

Reviewer #1 (Remarks to the Author):

This manuscript reported a novel ionic conductor, dynamic supramolecular ionic conductive elastomers (DSICE) via “phase-locked” strategy and explored the application in flexible iontronic devices. Besides, the performance of DSICE is very competitive compared with as-reported ionic conductors. Based on molecular engineering, the design strategy solves the multiple contradictions of the existing ionic conductors. I think this strategy is innovative and interesting, which is meaningful for the further design of the advanced ionic conductors. Therefore, this manuscript can be considered for publication in Nature Communications after minor revision. The detailed comments are listed below:

Response:

We appreciate the reviewer's valuable comments very much. We have seriously considered the reviewer's comments and revised the manuscript accordingly.

1. DSICE exhibits excellent mechanical properties, such as high stretchability, high strength and high toughness. How about its resilience?

Response:

Thank you very much for the valuable suggestions of the reviewers. The resilience of DSICE is good and time-dependent. For the resilience of DSICE, we have added more explicit descriptions in the revised manuscript as follows.

“Cyclic tensile tests at different strains (100%, 300%, 500%, 700%, 1000%) demonstrated the hysteresis loop became large with the increase of strain, indicating the dissipated strain energy during stretching (Fig. S15a). The six successive cyclic tensile curves at 300% strain showed an obvious hysteresis loop, but completely recovered to the original curve after a relatively short waiting time (~60 min) (Fig. S15b), suggesting DSICE exhibited the time-dependent self-recoverable energy-dissipation network.” (Line 264-269)

Fig. S15. Cyclic tensile tests of DSICE-20 under the deformation rate of 100 mm min^{-1} . (a) Single cyclic stress-strain curves at different strains (100%, 300%, 500%, 700%, 1000%) in successive stretching. (b) Consecutive cyclic tensile curves at a strain of 300%. After relaxing for 60 min, the cycle curve was nearly overlapped with the first cycle.

“Cyclic stress-strain tests were performed to evaluate the self-resilience of DSICE at a constant loading/unloading rate of 100 mm min^{-1} , DSICE-20 as a representative was shown in Fig. S15. Fig. S15a showed the single cyclic stress-strain curves at different strains (100%, 300%, 500%, 700%, 1000%) in the successive tensile process. It can be observed that DSICE-20 specimen exhibits a hysteresis loop and the larger the tensile strain, the more pronounced the hysteresis loop, indicating that DSICE-20 effectively dissipated strain energy caused by the breaking of dynamic bonds during stretching–retraction cycles at the different strains. Furthermore, Fig. S15b exhibited full deformation self-recovery performance of DSICE-20. When the DSICE-20 sample was loaded to 300% strain for six successive loading-unloading cycles, the first cycle displayed the large hysteresis loop and the hysteresis area of the following 2-6 cycles decreased remarkably. This could be attributed to the partially unrecovered dissociated dynamic bonds from the first cycle. However, after relaxing for 60 min, the loading-unloading cycle curve was almost overlapped to the first cycle curve, thus evidencing the self-recoverable energy-dissipation network of DSICE.” (Page 16 in the Supplementary Materials)

2. In the paper, the content of UPy is increased to 30% mol. Whether the increase of UPy will further improve the mechanical strength? It is known that the solubility of UPy is very poor. Will further increasing the content of UPy affect the reaction process?

Response:

We are very grateful for the reviewer's professional comments. When the content of UPy is increased to 40% mol, the mechanical strength does not enhance any longer, as shown in Fig R1. Owing to its rigid and planar aromatic structure, the UPy dimer in the hard phase tends to undergo self-assembly for further enhancement. On the other hand, influenced by the alicyclic isocyanates HMDI with large steric hinderance, the reaction of UPy was blocked and the aggregation of rigid segments would not further increase, thus resulting in the mechanical strength no longer enhancement.

Fig. R1. The stress-strain curve of DSE-4. DSE-4 refers to the dynamic supramolecular elastomers with the 40% mol UPy. DSE-4 exhibits mechanical performance with a stress up to 40.03 MPa and a stretchability of 1527.40%, which is slightly lower than DSE-3.

3. The conductive mechanism of DSICE is the dissociation of LiTFSI by the soft phase PTMEG and the movement of the chain segment to promote the transfer of lithium salt. So, will the ionic conductivity performance be better if the lithium salt is replaced with an ionic liquid? Please explore it.

Response:

We appreciate the reviewer's valuable suggestion. In this manuscript, LiTFSI chosen as conductive electrolyte salts is due to the weaker coordination interaction between Li^+ and ether oxygen bond in the PTMEG chain segments and the soft PTMEG chains are more in favor of ion transportation. However, ionic liquid (IL) with large volume ions has little interaction with DSE polymer chains, resulting in poor compatibility. Thus, IL in the DSE polymers does not exhibit ionic conductivity performance. We have replaced LiTFSI with the ionic liquid (IL) to explore its ion transport behavior.

[BMIm][NTf₂] was chosen as a representative type of ILs. Different from the ion conductive mechanism of LiTFSI in DSE elastomers, the IL [BMIm][NTf₂] is dispersed as droplets in DSE elastomers, forming a discontinuous ion transport path (Fig. R2a). Therefore, the IL inclusions in DSE elastomers does not exhibit ion transport feature but excellent dielectric properties (Fig. R2b).

Fig. R2. SEM images and dielectric property of the IL inclusions in DSE elastomers with 50 wt% [BMIm][NTf₂]. (a) SEM image of IL-inclusion composite elastomer, showing a uniform dispersion of near-spherical droplets with diameters of 5-10 um. (b) Dielectric constant of IL-inclusion composite elastomer and pure DSE elastomer versus testing frequency. The dielectric constant of IL-inclusion composite elastomer with 50 wt% [BMIm][NTf₂] is 2~3 times that of pure DSE elastomer.

4. Comparing Figure 2a and Figure 4e, the addition of LiTFSI leads to a certain decrease in the mechanical strength. What is the reason?

Response:

Thanks for the helpful suggestion raised by the referee. As suggested, we explained why the addition of lithium salts leads to a decrease in mechanical strength in line 229-231. Addition of LiTFSI salt causes a certain decrease in the mechanical properties of DSICE, which may be due to the large size of the TFSI anions interfering with chain packing and thus preventing aggregation of the UPy domains^{53,66}. For the reason that LiTFSI leads to a certain decrease in the mechanical strength, we have performed SAXS tests to further verify and added more explicit descriptions in the revised manuscript as:

“which was further verified via SAXS characterization (Fig. S14).” (Line 263)

Fig. S14. SAXS profile plots of DSICE-10~40. It demonstrates that the microphase separation structure of DSICE disappeared with the increase of LiTFSI content.

“The SAXS profile plots showed that the obvious microphase separation occurs when LiTFSI content is 10 wt.%, and the microphase separation disappears when the content continues to increase to more than 20 wt.%, indicating the increase of LiTFSI content hinders the accumulation of the hard phase chain domains. Hence, attenuation of hard phase chain segments stacking leads to the mechanical degradation of DSICE.”

(Page 15 in the supporting information)

References:

53. D. G. Mackanic, X. Yan, Q. Zhang, N. Matsuhisa, Z. Yu, Y. J. Jiang, T. Manika, J. Lopez, H. Yan, K. Liu, X. Chen, Y. Cui, Z. Bao, Decoupling of mechanical properties and ionic conductivity in supramolecular lithium ion conductors. *Nat. Commun.* **10**, 5384 (2019).
66. Y. Tominaga, K. Yamazaki, V. Nanthana, Effect of anions on lithium ion conduction in poly(ethylene carbonate)-based polymer electrolytes. *J. Electrochem. Soc.* **162**, A3133-A3136 (2015).

5. The author mentioned that DSICE has the potential as a solid-state battery application several times in the article. I recommend that the authors try it in this application.

Response:

Thank you very much for the valuable suggestion of the reviewer. We have made an attempt to apply DSICE in the solid-state battery application and achieved good results. DSICE-35 with high ionic conductivity and good mechanical properties was incorporated into LFP|SPE|Li coin cells to measure battery performance in an all-solid-state configuration. Fig. R3 demonstrated that the long-term cycling of DSICE-35 electrolyte at 60 °C and a rate of 0.1 C. The all solid-state cell could cycle stably with

a capacity retention of 91.2% after 50 cycles and an average coulombic efficiency of 96.3%. The battery performance demonstrated here is comparable to many literature examples of PEO solid electrolyte system. Therefore, the DSICE system has the potential application in the solid-state batteries, which we will continue to explore.

Fig. R3. Performance of DSICE-35 SPE in an all-solid-state battery with LFP|SPE|Li at 0.1 C at 60 °C. (a) Cycling stability and coulombic efficiency. (b) The charge-discharge curves.

Reviewer #2 (Remarks to the Author):

In this manuscript, the authors reported the design and construction of a novel ionic conductor, dynamic supramolecular ionic conductive elastomers (DSICE) via the promising “phase-locked” strategy. The strategy skillfully solves the conflicts between ionic conductivity, self-healing capacity and mechanical properties of existing ionic conductors and endows the well-designed DSICE with excellent comprehensive properties, which helps to expand the application scenarios of ionic conductors in stretchable sensors. Overall, the research was well designed and completed at a high level. Therefore, the manuscript would be suitable for publication in Nature Communications after the following questions are well addressed.

Response:

We are very grateful to the reviewer for the positive comments on our work. Our response to each question of the reviewer is as follows.

1. As we know, pure polymers cannot conduct ions as easy as liquid medium. However, as for the liquid-free ionic conductors in this work, the DSICE exhibited high ionic conductivity. What is the reason behind the good performance? It's better to highlight this aspect because it is very important for potential or even practical applications in various fields including flexible sensors and all solid state batteries.

Response:

Thanks for the reviewer's valuable suggestion. Inspired by PEO-based polymer electrolytes, lithium salts are dissociated into lithium ions (Li^+) and the corresponding negative ions by the polar atom O in the PEO backbone, and dissociated Li^+ hop along with the polymer chain segments movement, achieving the ions transportation. Therefore, the transportation of ions in the polymer is correlated to the degree of Li^+ dissociation and coordination. Unfortunately, the strong O- Li^+ interaction in EO-based polymers leads to large activation energy for ion motion. (*Faraday Trans.* **1993**, 89, 3187). The chemical component of PTMEG in this manuscript is similar to PEO, made up of CH_2 and O. However, the repeating units of PTMEG include four carbons and one oxygen atom, compared with that of PEO made up by two carbons and one oxygen atom, the oxygen concentration of PTMEG in the vicinity of lithium ion is lower.

Therefore, the O-Li⁺ coordination effect in the PTMEG backbone is relatively weak. In addition, PTMEG-based backbone has a strong de-shielding effect on Li⁺, developing a loose O-Li⁺ coordination interaction, which effectively improve ion dissociation, facilitate ion transportation, and enhance ionic conductivity (*Adv. Energy Mater.* **2018**, 8, 1800703; *Chem. Eng. J.* **2021**, 423, 130209). Therefore, the higher ionic conductivity of DSICE was explained in more detail, as depicted in the revised manuscript.

“in which the loosely coordinating O-Li⁺ interaction and lower activation energy for ion transport can contribute to higher ion conductivity^{57,58}.” (Line 90)

“The high ionic conductivity mainly owes to the loosely Li⁺-O coordinating interaction and lower activation energy for ion transport of PTMEG soft polymer chains (Fig. 4b).” (Line 236-238)

Fig. 4. (b) Schematic illustration of Li⁺ transporting in the PTMEG chains with loose coordination structure.

2. The authors claimed that “Li⁺ dissociation environment in the DSICE materials derive from the soft PTMEG segments and the UPy groups do not interfere in the Li⁺ dissociation and transport” in the Lines 223 and 224.

It is understood that highly polar atoms are conducive to the dissociation of lithium salts, and N and S elements contained in the designed structure may also contribute to the dissociation of lithium salts. Please discuss whether it is possible to happen.

Response:

The reviewers’ suggestions at this point are precious. Ion transport in polymers consists of two processes: the dissociation of lithium salts and the dissociated ions transport along with the free-moving chain segments (*J. Mater. Chem. A*, **2015**, 3,

19218-19253). The dissociation of lithium salts depends on the polar groups (C-O, C=O, -NH, -S, etc.) in the polymer segments (*Polymer*, **2010**, 51, 2864-2871), so the N and S elements in the chemical structure of DSICE could contribute to the dissociation of lithium salts, generating more dissociative ions. On the other hand, PTMEG soft chain segments movement facilitates ion transport, thus imparting higher ionic conductivity. For the contribution of N and S elements of DSICE structure to the dissociation of lithium salts, we have conducted solid-state ^7Li NMR to study the chemical environment of Li^+ . The ^7Li NMR spectra was shown in Fig. 4d. And the relevant explanations have been depicted in the revised manuscript.

“Ion transport mechanism was further investigated. Solid-state ^7Li NMR was employed to study the dissociation environment. Fig. 4c showed the ^7Li NMR traces of DSICE (DSE-0, DSE-1, DSE-2, DSE-3 with 35 wt.% LiTFSI, respectively). It can be observed that the ^7Li shifts downfield with increasing amounts of UPy (the increase of highly polar N elements), suggesting that the increase of UPy amounts leads to ion pair dissociation, generating the more dissociated Li^+ and the counterparts⁶⁵.” (Line 240-244)

Fig. 4. (c) ^7Li NMR traces of DSICE (DSE-0, DSE-1, DSE-2, DSE-3 with 35 wt.% LiTFSI).

References:

65. B. Qiao, G. M. Leverick, W. Zhao, A. H. Flood, J. A. Johnson, Y. Shao-Horn, Supramolecular regulation of anions enhances conductivity and transference number of lithium in liquid electrolytes. *J. Am. Chem. Soc.* **140**, 10932-10936 (2018).

3. The pictures in Figs. 3c and 6a should have scale bars. Besides, there are several mistakes in the main text. For example: please change “combing” into “combining” in

line 237; please change “.” into “,” in line 261; please change “amount” into “amounts” in line 208.

Response:

We are so sorry for the details in the manuscript. The scale bars in the Figs. 3c and 6a have been added into the revised manuscript. Also, we have revised the spelling, punctuation and grammar in line 237, line 293 and line 230. In addition, we have gone through the entire manuscript to avoid similar problems carefully. All of the changes were highlighted in yellow in the revised manuscript.

Fig. 3 (h) Optical microscopy images of the scratched and healed DSE-2 film. Healed condition: 12 h at ambient environment. The scale bar: 70 μm.

Fig. 6 (a) Visually demonstration of DSICE as conducting substrate with excellent self-healing capacity to extend the service life. From left to right: A “heart-shaped” pattern with 18 LEDs was placed on the DSICE-30 substrate; LEDs were lit in an electric field; DSICE-30 sample was cut from the middle of the “heart-shaped” pattern and LEDs were out; after self-healed for 30 min, LEDs were lit again.

Due to changes in the content of the article, the word of combining in line 237 has changed into “To sum up,” (Line 310)

“DSICE were endowed with excellent autonomous self-healing capability,” (Line 294)

“dissolving different amounts of LiTFSI from 10 wt.% to 40 wt.% into the DSE polymers” (Line 231)

4. The authors claimed that “as the amount of LiTFSI increases, the mechanical strength decreases and the stretch strain increases. This is due to the loose polymer chains

stacking caused by the increasing TFSI anions”. Are there any evidences to support the hypothesis?

Response:

We thank the referee for the helpful suggestion. As suggested, we have performed SAXS tests to further verify and added more explicit descriptions in the revised manuscript as:

“which was further verified via SAXS characterization (Fig. S14).” (Line 263)

Fig. S14. SAXS profile plots of DSICE-10~40. It demonstrates that the microphase separation structure of DSICE disappeared with the increase of LiTFSI content.

“The SAXS profile plots showed that the obvious microphase separation occurs when LiTFSI content is 10 wt.%, and the microphase separation disappears when the content continues to increase to more than 20 wt.%, indicating the increase of LiTFSI content hinders the accumulation of the hard phase chain domains. Hence, attenuation of hard phase chain segments stacking leads to the mechanical degradation of DSICE.”

(Page 15 in the Supplementary Materials)

5. The demo of flexible and stretchable touch sensor is very impressive, which takes full advantages of the merits of DSICE including conductivity, flexibility, stretchability and mechanical robustness. However, for the reviewer and potential readers, it's not clear that what's the working principle or mechanism of DSICE-based touch sensor. It is suggested to provide more details on this.

Response:

We appreciate the reviewer's valuable advice. DSICE with ionic conductive feature

have a larger response at different stimulus states (original, touched, stretched, stretched and touched) in the whole frequency range. As advised, we have added more detailed descriptions on the DSICE-based touch sensor in the revised manuscript, as follows:

“DSICE as ionic conductors has the frequency-dependent feature. Therefore, we used the impedance spectra and impedance complex plane to recognize different stimulus of the touch sensors.” (Line 325-326)

“Because of the frequency-dependent feature of DSICE, the impedance spectra can change at different stimulus states over the whole frequency range (1 MHz to 0.1 Hz).” (Line 330-331)

“When a single frequency is fixed, we can accomplish real-time monitoring, getting both Z' and Z'' in a single acquisition. In this way, different stimuli will appear in different region in the two-dimensional impedance complex plane. Actually, the whole frequency range (1 MHz to 100 Hz) impedance spectra was to find an optimal frequency which will better distinguish the different stimuli, then, we fix the single optimal frequency to detect the different stimulus.” (Line 341-345)

6. The synergistic effect between S-S bond and UPy H-bonds play an important roles in both conductivity and mechanical performance. However, there is a lack of deep discussion about the synergy. How it works or how they work together to boost the performance of the DSICEs.

Response:

We deeply thank the reviewer for this suggestion. First, we explain that the synergistic effect between S-S bonds and UPy H-bonds play an important role in both self-healing capacity and mechanical performance. In this manuscript, the introduction of multiple dynamic bonds including disulfide metathesis (S-S), strong crosslinking H-bonds (UPy-UPy) and weak crosslinking H-bonds (urethane-urethane, urea-urea, or urea-urethane) could spontaneously forming a dynamic supramolecular polymer network (Fig. 1a). In this dynamic supramolecular polymer network, the disulfide bonds mainly contribute to the self-healing capacity, the strong crosslinking H-bonds primarily endow physical crosslinking for mechanical enhancement while weak H-bonds as sacrificial bonds can dissipate strain energy, increase toughness and retain the

basic structure via efficient reversible bonds rupture and recombination. It is the synergistic effect of these multiple dynamic bonds that imparts the DSICE with superior mechanical properties, excellent self-healing capability as well as recyclability.

For the synergy effect on the self-healing capacity and mechanical performance, we have discussed in-depth in the revised manuscript below.

“while the synergistic effect of the dynamic disulfide metathesis (S-S) and stronger supramolecular quadruple hydrogen bonds (H-bonds) in the hard phase domains was used to regulating the self-healing capacity and mechanical properties.” (Line 83)

“in which the disulfide bonds mainly contribute to the self-healing capacity, the strong crosslinking H-bonds is used for mechanical enhancement while weak H-bonds as sacrificial bonds can dissipate strain energy, increase toughness via efficient reversible bonds rupture and recombination,” (Line 95-98)

“Achieving both mechanical and self-healing properties through the synergistic interaction between HEDS and UPy.” (Line 161-162)

7. As for the self-healing of DSICE, apart from the explanations of dynamic H-bonding and S-S bond, the reviewer thinks the Li-O dipole interactions are also contributable to the fast self-healing. In fact, it is an extra bonus of the smart design.

Response:

Thanks to the reviewer for this precious perspective and guidance. Compared Fig. 3i and Fig. 5b in the manuscript, the self-healing capability of DSICE with Li-O dipole interactions is much better than that of DSE without Li-O dipole interactions. In addition, we have investigated ionic conductors with Li-O dipole interactions contributing to the fast self-healing (*Adv. Mater.* **2021**, 2006111; *Adv. Sci.*, **2021**, 8, 210139; *NPG Asia Mater.* **2022**, 14, 10; *Science* **2021**, 374 (6564) 150-151; *Nat. Commun.* **2022**, 13, 2279). Hence, we agree with the review’s point of view. We have added the suggestion raised by the review in the revised manuscript.

“which ascribed to the critical contribution of the reversible nature of dynamic S-S bonds, supramolecular H-bond motifs and extra Li-O dipole interactions^{29,62}.” (Line 296)

References:

29. B. Yiming, Y. Han, Z. Han, X. Zhang, Y. Li, W. Lian, M. Zhang, J. Yin, T. Sun, Z. Wu, T. Li, J. Fu, Z. Jia, S. Qu, A mechanically robust and versatile liquid-free ionic conductive elastomer. *Adv. Mater.* **33**, e2006111 (2021).

62. M. Li, L. Chen, Y. Li, X. Dai, Z. Jin, Y. Zhang, W. feng, L.-T. Yan, Y. Cao, C. Wang, Superstretchable, yet stiff, fatigue-resistant ligament-like elastomers. *Nat. Commun.* **13**, 2279 (2022).

Reviewer #3 (Remarks to the Author):

Over comment: The authors here report on a highly stretchable, self-healable ionically conducting elastomer. Certainly, those types of materials can find a plethora of applications in intronic devices. Hence, there is a rich research background existing on those types of elastomers.

Response:

We thank the reviewer for the careful review and the constructive comments. In this study, we aimed to resolve the conflicts among ionic conductivity, self-healing capability, and mechanical compatibility (strength, stretchability and toughness) in the existing ionic conductors and impart DSICE with outstanding comprehensive performance via elaborately designing the chemical structures. Furthermore, we have provided additional experimental data, enriched the intronic touch sensor devices and reorganized our discussion in more detail in the revised version. Responses to each issue are given as follows.

Comment 1: In this paper, the authors show high levels of stretchability in their thermoplastic polyurethane based system. However, as typical for such systems, this behavior is mostly viscoelastic/plastic in nature, with little reversibility. In fact, the authors do not show restoring capabilities of their material when stretched to >300% tensile.

Response:

We appreciate the reviewer's detailed and critical comments. As is well-known, the thermoplastic polyurethane based system is mostly viscoelastic/plastic in nature, which depends on their linear molecular structure without chemical crosslinking but physical cross-linking via hydrogen bonds between molecular chains. Because the reconstruction of the broken hydrogen bonds as reversible structures usually takes minutes or hours. Thus, for the PU system, its restoring capability is time-dependent reversibility rather than little reversibility.

From the perspective of polymer physics, the viscoelastic behavior/plastic of PU based polymers is directly related to their underlying molecular chain's movement. In the unstretched state, the long polymer chains are interconnected or physical

entanglements, forming a dynamic polymer network. When initially stretched, the molecular links or segments begin to vibrate, and once the stretch is removed, the vibration stops instantly, exhibiting fully reversible elastic recovery. When further stretched, the interconnected, entangled and physical crosslinked polymer chains are unfolded, and when released, the unfolded polymer chains and the physical crosslinked point would refold, reorganize and reconstruct over time to keep its lowest energy state, exhibiting time-dependent reversible elastic recovery. When stretched to the maximum extent, the fully unfolded polymer long chains slip and stagger, eventually break, which would lead to the irreversible recovery. Therefore, the irreversibility of the PU based polymer only occurs in the fully unfolded polymer long chains slippage (very large deformation) stage. When the tensile deformation is at certain stage (the interconnected, entangled and physical crosslinked polymer chains are unfolded), the time-dependent reversible recovery occurs. Therefore, we have added the detailed explanation on the chain segments change situation along stretching in the revised manuscript.

“Interestingly, both DSE materials exhibited “J-shaped” stress-strain curves that was strain-dependent mechanical responses. In the unstretched state, the interchain loops lead to the folding of the polymer backbones. At low strains, a linear increase of stress occurs, corresponding to the vibration of the molecular links or segments. When keep stretching, the stress increases slowly with increasing strain, corresponding to the extensive soft chain segments, the breakage of dynamic S-S bonds and weak H-bonds. And further stretched, significant strain hardening occurs in the third stage, corresponding to the enhancement of UPy quadruple H-bonds and the unfolding and sliding of the polymer backbones. (Fig. 1a).” (Line 166-172)

As the reviewer suggested, we have measured the restoring capabilities of this materials when stretched to 300% and 500% tensile. In the revised manuscript, “Fig. 3d displayed the good deformation recovery of DSE-3 from an elongation of 300% and the restoring capability is time-dependent. In addition, six successive loading-unloading cycles tests were performed to further evaluate the self-resilience of DSE-3 at a strain of 300%, as shown in Fig. 3e. The cyclic curves displayed a pronounced hysteresis and the 50% residual strain. Besides, the first cycle displayed the larger

hysteresis loop and the hysteresis area of the following 2-6 cycles decreased remarkably, which could be attributed to the partially unrecovered dissociated dynamic bonds from the first cycle. Furthermore, after relaxing for 10 min, the loading-unloading cycle curve was almost overlapped to the first cycle curve, exhibiting the full recovery of the hysteresis area and the appearance of residual strain. This time-dependent self-recovery property mainly depends on the reversible dissociation/reassociation of relatively weak H-bonds and S-S bonds in the dynamic hard domains.” (Line 189-198)

Fig. 3. (d) Digital images for DSE-3 film that can restore to its original length after being stretched to 300% strain. **(e)** Consecutive cyclic tensile curves of DSE-3 at a strain of 300%. After relaxing for 10 min, the cycle curve was fully overlapped with the first cycle, indicating the full self-recoverability of DSE.

“The same is true for the DSE-3 loaded to 500% strain (Fig. S9).” (Line 198-199)

Fig. S9. Consecutive cyclic tensile curves of DSE-3 at a strain of 500%. After relaxing for 30 min, the cycle curve was nearly overlapped with the first cycle, indicating the full self-recovery of the hysteresis area and the residual strain.

Comment 2-1: Sadly, this is somewhat typical in this research field, however placing strong emphasis on large but irreversible stretchability holds little promise for practical applications.

Response:

Thanks to the reviewers for their concerns about this point. In the recent decades, various engineering applications, ranging from flexible electronic/ionotronic devices to soft robotics, have increased the demand for versatile and high-performance soft materials, which has received widespread attention in this research field. In many of these applications, soft polymeric materials (elastomers) can have the required physicochemical and mechanical properties but often lack the necessary toughness (the ability to dissipate much energy and deform without breaking) to maintain functional unfailure when subjected to large deformations.

The ability of soft polymeric materials to achieve desired toughness is directly related to their underlying molecular structure. Elastomers consist of long polymer chains that are interconnected, either by chemical cross-links or physical entanglements, and form a network. Most tough elastomers are reinforced by introducing sacrificial structures (such as hydrogen bonds, dynamic bonds, metal coordination bonds, ionic bonds etc.) that can dissipate input energy (*Prog. Polym. Sci.* **2018**, 80, 39; *Nat. Rev. Mater.* **2021**, 6, 421; *Mater. Horiz.* **2020**, 7, 2882). By using sacrificial structures, of which the reconstruction usually takes minutes or hours, as the energy dissipation mechanism, tough elastomers with a high extent of structural and mechanical recovery can be realized within a certain amount of time. Therefore, soft polymeric materials that can withstand large deformations hold great promise for practical applications. The ability to sustain large deformations is just the embodiment of high toughness that can resist outside break. And this large deformation is not irreversible but time-dependent reversible. On the other hand, it is worth noting that the fast deformation and recovery at the low strain (10-50%) is desirable for flexible devices.

Comment 2-2: Instead, a stronger focus on characterizing the "moderate" strain region (here up to 300%, one curve up to 500% in the SI) more exhaustively would be required. As shown in figure 3b, the material exhibits strong hysteresis effects (which the authors

acknowledge and favor as this is associated with energy dissipation and "toughness") of the material, but also demonstrates considerable irreversibility of deformation (plastic deformation) of 50% strain. This should be discussed in more detail.

Response:

We appreciate the reviewer's critical comment. As the reviewer pointed out, we have further added the mechanical and recovery characterization of the material and discussed the internal mechanism in more detail.

“Fig. 3c showed the single cyclic stress-strain curves of DSE-3 at different strains (100%, 300%, 500%, 700%) in the successive tensile process. It can be observed that DSE-3 specimen exhibits a hysteresis loop and the larger the tensile strain, the more pronounced the hysteresis loop, which indicated that DSE-3 effectively dissipated strain energy caused by the breaking of dynamic bonds during stretching–retraction cycles at the different strains. Fig. 3d displayed the good deformation recovery of DSE-3 from an elongation of 300% and the restoring capability is time-dependent. In addition, six successive loading-unloading cycles tests were performed to further evaluate the self-resilience of DSE-3 at a strain of 300%, as shown in Fig. 3e. The cyclic curves displayed a pronounced hysteresis and the 50% residual strain. Besides, the first cycle displayed the larger hysteresis loop and the hysteresis area of the following 2-6 cycles decreased remarkably, which could be attributed to the partially unrecovered dissociated dynamic bonds from the first cycle. Furthermore, after relaxing for 10 min, the loading-unloading cycle curve was almost overlapped to the first cycle curve, exhibiting the full recovery of the hysteresis area and the appearance of residual strain (Fig. 3d). This time-dependent self-recovery property mainly depends on the reversible dissociation/reassociation of relatively weak H-bonds and S-S bonds in the dynamic hard domains. The same is true for the DSE-3 loaded to 500% strain.” (Line 183-199)

Comment 2-3: 5 stress-strain cycles are not sufficient to study the long-term mechanical behavior of this material and compare it to other systems like hydrogels with ionic liquids. In general, materials reliability and long-term performance is not assessed here at all.

Response:

We thank the referees for their valuable suggestions, which is helpful to improve the quality of our work. 5 cyclic tensile tests were performed to assess the self-recovery property (energy dissipation) of this material (*Adv. Mater.* **2021**, 2101498). In our designed material system, multiple dynamic bonds were introduced into the hard phase domains to effectively resolve the conflict between self-healing, toughness and mechanical strength in the traditional elastomers. For hydrogels or organogels with ionic liquids, they consist of polymeric networks and water, organic solvent or ionic liquids. Polymeric networks usually determine the mechanical properties (toughness, stretchability, and strength). Covalently crosslinked polymeric networks can effectively enhance the mechanical robustness but impair the stretchability and toughness because of its intrinsic irreversibility. Reversible noncovalent bonds (hydrogen bond, dynamic chemical bonds, and ionic interaction etc.) as sacrificial bonds are expected to simultaneously improve the stretchability and toughness of hydrogels or organogels with ionic liquids (*Adv. Mater.* **2016**, 28, 4884-4890; *Adv. Mater.* **2017**, 29, 1700759; *Science* **2021**, 374, 6564). Therefore, whether it is a gel or an elastomer with sacrificial bonds, it will show a hysteresis effect and time-dependent recovery process (*Nature* **2012**, 489, 133-136; *Nat. Mater.* **2013**, 12, 932-937; *Adv. Mater.* **2016**, 28, 4678-4683).

As suggested, the reliability and long-term performance of this material was conducted in the revised manuscript. “In this work, the DSICE materials exhibited favorable reliability and long-term performance, as shown in Fig. S17. DSICE-20 as a representative can undergo up to 600 cycles at a higher strain of 100% with a certain stress relaxation (Fig. S17a), which stems from the break and recombination of the dynamic bonds. It is worth noting that the fast deformation and recovery at the low strain (10-50%) is desirable for flexible devices. Thus, when the strain decreased to 30%, the fresh DSICE-20 and one stored for 3 months cycled up to 375 cycles stably with negligible stress relaxation (Fig. S17b).” (Line 273-279)

Fig. S17. The reliability and long-term performance of DSICE. (a) Cyclic durability of DSICE-20 at 100% strain for 600 cycles (12000 s). (b) Cyclic stability of the fresh DSICE-20 and one stored for 3-month at 30% strain for 375 cycles (1500 s).

Comment 3: In addition, the claim on "superior toughness" is not at all substantiated. I assume the authors refer to fracture toughness of the material. However, they simply integrate the stress-strain curve up to rupture of their elastomer (Figure S8). A more proper investigation on crack propagation/crack opening and the associated energies is required. I refer here to the seminal work of Zhigang Suo's group on the topic of fracture toughness in soft materials.

Response:

Thanks to the reviewer for the valuable comments and helpful suggestions. We agree with the reviewer's comment. The "superior toughness" in this manuscript refers to the fracture toughness, which is calculated by the integrate area of the stress-strain curve, indicating how much energy the material can absorb (*Adv. Mater.* **2021**, 2101498; *Adv. Mater.* **2018**, 30, 170514; *Adv. Funct. Mater.* **2019**, 1907109; *Nat. Commun.* **2022**, 13, 2279). We have added the concept of toughness in the revised manuscript.

"Toughness, which is related to energy dissipation, has positively correlation with mechanical strength and stretchability." (Line 179-180)

As the reviewer suggested, we have investigated and studied on the crack propagation/crack opening and the associated fracture energy that is the work required to advance a crack by unit area. (*Science* **2021**, 374 (6654), 212-216; *Extreme Mech. Lett.* **2021**, 48, 101434; *J. Mech. Phys. Solids* **2020**, 134, 103751; *Macromol. Rapid Comm.* **2019**, 40(8), 1800883; *Soft Matter* **2018**, 14, 3563-3571; *Extreme Mech. Lett.*

2017, 10, 24-31; *Eng. Fract. Mech.* 2018, 187, 74-93; *J. Appl. Mech.* 2020, 87, 031002; *Nat. Commun.* 2022, 13, 2279). Furthermore, we have performed the fracture energy tests. Thanks to the remarkable energy dissipation capability, the pre-cracked DSE and DSICE materials exhibit superb crack tolerance.

“To illustrate the crack resistance of DSCE materials, fracture energy was measured to quantitatively assess the crack tolerance via the well-established method for rubber, Rivlin-Thomas pure shear test⁶⁰. To observe the crack propagation, two sets of large DSE-3 samples with width of 500 mm and height of 100 mm were prepared, of which one has 200 mm precut crack (the size of crack was twice of the height of the sample), and the other one without crack^{61,62}. As shown in Fig. 3f and Movie S1, the precut crack was observed to be obvious blunt during stretching. The crack initiated at the front of the notch, propagated in the longitudinal direction rather than the transverse direction and eventually fractured. Fig. 3g showed the pre-damaged DSE-3 sample with 200 mm precut crack can be stretched to approximately 4.5 times its original height. The ultimate stress and the stretch ratio reach as high as 3.82 MPa and 5.18, respectively. The fracture energy⁶¹ of the sample is calculated to be as large as 201.29 kJ m⁻². The results above demonstrate its exceptional fracture energy, which is caused by the special interior structure of the dynamic hard phase domains.” (Line 200-210)

Fig. 3. (f) Photographs of the intact DSE-3 sample and one with the precut crack stretched to 3 times its original stretching at a tensile rate of 100 mm min⁻¹. (g) Stress–strain of the intact DSE-3 sample and one with the precut crack (gauge length: 10 mm). The fracture energy (Γ) is calculated by the formula: $\Gamma = HW(\lambda_c)$.

References:

60. R. S. Rivlin, A. G. Thomas, Rupture of rubber. I. Characteristic energy for tearing.

J. Polym. Sci. **10**, 291–318 (1953).

61. Y. Zhou, W. Zhang, J. Hu, J. Tang, C. Jin, Z. Suo, T. Lu, The stiffness-threshold conflict in polymer networks and a resolution. *J. Appl. Mech.* **87**, 031002-1 (2020).

62. M. Li, L. Chen, Y. Li, X. Dai, Z. Jin, Y. Zhang, W. feng, L.-T. Yan, Y. Cao, C. Wang, Superstretchable, yet stiff, fatigue-resistant ligament-like elastomers. *Nat. Commun.* **13**, 2279 (2022).

“Rivlin-Thomas pure shear tests were performed to further assess the fracture energy of DSICE materials (Fig. S16)^{61,62}.” (Line 269-270)

Fig. S16. Stress-strain of the intact DSICE-20 and DSICE-30 samples and one with the precut crack (gauge length: 10 mm). The fracture energy (Γ) is calculated by the formula: $\Gamma = HW(\lambda_c)$.

“The pre-damaged DSICE-20 sample with 200 mm precut crack could be stretched to approximately 3.7 times and showed a fracture energy of 67041.68 J m⁻². And that of the pre-damaged DSICE-30 showed 23955.94 J m⁻², indicating the DSICE materials have favorable crack tolerance and the efficient energy dissipation.” (Page 17 in the Supplementary Materials)

Comment 4: The re-usability part (Figure 5) lacks characterization, and essentially only consists of a couple photographs. Where are mechanical tests on re-processed material samples? Is it possible to repeat this process? Some data at least should be shown to support the argument.

Response:

We agree with the reviewer’s suggestion. We have conducted the mechanical stress-strain curves tests of reprocessed DSICE specimens, as shown in Fig. S21 and depicted in line 305-307.

“As shown in Fig. S21, the reprocessed DSICE-35 showed slightly decreased

stretching stress, which is attributed to the insufficient crosslinking of the UPy units after the recycling process.” (Line 308-310)

Fig. S21. Typical stress-stain curves of DSICE-35 after three recycling.

Comment 5-1: Finally, the demonstrations of the material as ionic conductor are very rudimental at best. The demo with the LEDs is in the presented form not understandable. There seems to be some LEDs placed onto the ionic conductor, then a "electric field" is applied. How does this work in detail? Are there structured conductive traces somewhere, or is there just a high AC field applied? Not very exhaustive description of the experiment.

Response:

Thanks for the reviewer’s helpful suggestion. To demonstrate the high ionic conductivity and excellent self-healing capacity of our designed DSICE materials visually, the demo with the LEDs over the flexible DSICE substrate was designed. If the ionic conductivity is lower, the internal resistant of the formed circuit is larger, which results in the LEDs over the ionic conductor substrate not be lighting up. On the other hand, the excellent self-healing capacity of DSICE is beneficial to improve the material reliability and prolong the service life. Therefore, DSICE as a novel ionic conductor without structured conductive traces is very favorable for the flexible conductive substrate application.

As suggested, we have updated a more detailed explanation on how the LEDs onto the ionic conductor works in the revised manuscript. More details are as following.

“As a proof-of -concept, the demo with the LEDs over the flexible DSICE substrate

was designed, as shown in Fig. 6a.” (Line 315-316)

“The LED lights on the DSICE-30 substrate can form a conductive circuit with an AC 220 V electric field. Because of high ionic conductivity, the “heart-shaped” pattern with LED lights can be entirely lit.” (Line 317-319)

Comment 5-2: Then, the "touch panel" is quite undercharacterised as well. No photographs or proper touch measurements are given. Nor proper mechanical characterization of that panel. Compared to what is out there in literature on ionic touch panels, this demo falls way short. Quite some reworking of the demo section is required to convincingly show the feasibility of this material in practical applications, and its superiority compared to other approaches.

Response:

We appreciate the reviewer's constructive comment. The “touch sensor” in the manuscript is a relatively small minority impedance sensor. DSICE with ionic conductive feature have a larger response at different stimulus states (original, touched, stretched, stretched and touched) in the whole frequency range. As suggested, the photographs of DSICE-30 at four different stimulus (original, touched, stretched, stretched and touched) have added in the Supporting information. In addition, because DSICE-based touch sensor has response to the stretched and touched state. Therefore, we conducted the impedance change when stretched to different tensile elongation (Fig. S23 and Movie S3) and the impedance change at different touched stimuli when stretched to a certain strain (Fig. S24 and Movie S4). We have added the more information and explanation on the independent touch sensor in the revise manuscript and supporting information.

“DSICE as ionic conductors has the frequency-dependent feature. Therefore, we used the impedance spectra and impedance complex plane to recognize different stimulus of the touch sensors.” (Line 325-326)

“Because of the frequency-dependent feature of DSICE, the impedance spectra can change at different stimulus states over the whole frequency range (1 MHz to 0.1 Hz).” (Line 330-331)

“When a single frequency is fixed, we can accomplish real-time monitoring, getting

both Z' and Z'' in a single acquisition. In this way, different stimuli will appear in different region in the two-dimensional impedance complex plane. Actually, the whole frequency range (1 MHz to 100 Hz) impedance spectra was to find an optimal frequency which will better distinguish the different stimuli, then, we fix the single optimal frequency to detect the different stimulus.” (Line 341-345)

“As shown in Fig. 6c, 6d and Fig. S22,” (Line 332)

Fig. S22. The actual test photographs of DSICE-30 at four different stimulus (original, touched, stretched, stretched and touched). (a), (b), (c), (d) corresponds to original, touched, stretched, stretched and touched, respectively.

Furthermore, to enrich the touch sensor application of DSICE material, we present a capacitance touch sensor. The capacitance touch sensor aims to detect the present position of a finger in the panel. The proximity of the finger causes a drop in the capacitance between the electrodes. The drop capacitance occurs because the electric field is increasingly directed towards the finger, thus reducing the shared charge between the two electrodes. This type of proximity sensing works well in rigid indium tin oxide sensor devices and is known as mutual capacitance sensing in our daily touch devices. However, indium tin oxide touch sensor is rigid and brittle, which is not suitable for the promising flexible and stretchable electronic devices. DSICE with high ionic conductivity, excellent mechanical properties (flexible, stretchability, strength, toughness), autonomous self-healing capacity and high transparency is expected to be a candidate for future flexible electronic touch devices. Based on this, we prepared a proof-of-concept 4×4 cross-grid sensor array with a 10-mm pitch in our lab.

“Furthermore, we present a capacitance touch sensor to enrich the application of DSICE material. A proof-of-concept 4×4 cross-grid sensor array with a 10-mm pitch was demonstrated. The capacitance touch sensor was created by a cross-grid array of

transparent DSICE-30 electrodes capacitively coupled through a thin silicon film (100 μm), as shown in Fig. 6g. Fig. 6h exhibited the work principle of the capacitance touch sensor. In this implementation, each array element is composed of two electrodes (one is red and the other blue) separated by a dielectric layer. The finger acts as the third electrode, which capacitively couples to one electrode (blue), as represented by the variable capacitor C_F , thus reducing the coupling between two electrodes C_M ⁶⁸. The proximity of a finger causes a drop in the capacitance between (C_M) the electrodes. The drop capacitance occurs because the electric field is increasingly directed towards the finger, thus reducing the shared charge between the two electrodes. On the other hand, the direction position of a finger can be detected via the observed capacitance change. Each combination of row and column electrodes was scanned sequentially, and the capacitance changes of all elements were determined to create a map, as shown in Fig. 6i, 6j and Fig. S25 and Movie S5. There is a finger at the intersection of row 2 and column 2, that is the array element 1, which was connected with the LCR instrument to measure the 43.6% decrease in capacitance. The two vertically adjacent elements 2 and 3 yield a more 34.6% and 40.2% decrease, respectively, indicating the direction position of a finger.” (Line 349-364)

Fig. 6. (g) The schematic of the DSCIE capacitance touch sensor with a 4×4 cross-grid array. (h) the work principle of the capacitance touch sensor (i) and (j) The Map and actual test photographs showing the localized change in capacitance due to a touch by a finger.

Fig. S25. The initial capacitance value (untouched) and the capacitance value of different touch points by a finger. (a) The initial capacitance value (untouched). (b) The capacitance value of the finger touching the array element 1. (c) The finger touching the array element 2. (d) the finger touching the array element 3.

“In general, whether it is the relatively small minority impedance sensor or the pervasive capacitive sensor, DSICE exhibited the excellent touch sensing feature. Besides, its outstanding comprehensive performance provides great potential in the future of flexible electronics and soft robotics application.” (Line 364-367)

References:

68. M. S. Sarwar, Y. Dobashi, C. Preston, J. K. M. Wyss, S. Mirabbasi, J. D. W. Madden, Bend, stretch, and touch: Locating a finger on an actively deformed transparent sensor array, *Sci. Adv.* **3**, e1602200 (2017).

Comment 6: In summary, while the material itself is interesting, and the synthesis seems to be done well, this paper lacks essential information and needs major overhaul. At the current stage, a more specialized chemistry journal would seem a better fit.

Response:

We are very grateful to the reviewer’s positive comments on our material itself and its synthesis. In addition, we deeply thank to the reviewer for the valuable and crucial suggestions, which is very helpful to improve the quality of our manuscript. We have seriously considered the reviewer’s suggestions and revised the manuscript accordingly. We hope that the revised manuscript can be considered for publication in *Nature Communications*.

Reviewers' Comments:

Reviewer #1:

Remarks to the Author:

I would like to support the acceptance of this manuscript

Reviewer #2:

Remarks to the Author:

All concerns raised by the reviewers have been well addressed and thus the manuscript is now at a great state. Please consider to publish as is.

Reviewer #3:

Remarks to the Author:

In their revision, the authors have addressed several comments of the reviewers. Some are clear and nicely answered, for example the information on cyclic mechanical load is now relatively complete.

Some other experiments need however further clarification.

For once, i could not completely deduce from the author's reply and the revised manuscript, how in detail the fracture toughness tests were performed and how the values for Gamma were calculated. In particular, it is not clear to me how the crack propagation length $\lambda(c)$ or L_c was determined. Typically, the notched sample is stretched until the pre-formed crack propagates, and this is then taken as L_c ($\lambda(c)$) and serves as upper limit for the integrated area in the stress-strain curve of the un-notched sample. (Like in the works of Suo et al.).

Now, to my understanding, from the text of the authors, it appears the notched samples were stretched until they break, and this was taken as $\lambda(c)$. Such an approach would significantly over-estimate the fracture toughness of the material, and the reported values that exceed 200 kJ/m² seem much too large even for Polyurethanes. An exact description of the measurement procedure is required and would be much appreciated. Please clearly define all variables and formulas used. Including information on standard deviation and number of samples measured.

As for the demos, i am sorry, but i still don't fully understand how the heart shaped LED circuit works. Now, the authors state that the LEDs are "powered by an 220V AC electric field), there is however still no information on how the LEDs are contacted. Are they simply placed on top of a strip of ionic conductor? in this case, does the required voltage difference for driving the LEDs simply stem from the difference in resistance between the legs of the LED? That seems a highly inefficient and unpractical approach to power LEDs, even with ionic conductors. What then is the driving frequency?

There is still not much detailed information of the above kind in the methods section.

In addition, though the touch panel demos have been improved, i find it very difficult to make out the readings on the photographed impedance analyzers in the small pictures. For example, figure 6c does not seem to have much value other than showing that the impedance analyzer is switched on.

The same goes for figure S22, here one can also barely make out the readings on the photographed screens of the impedance analyzer. Some of this data is plotted in figure 6, right? How would one discriminate touch positions from the impedance spectra? it seems position detection with this kind of sensor seems challenging.

If the matrix-based touch demo consists of cross-bars of ionic conductors, separated by a deformable dielectric, then pressing them should also change capacity, not only touching with a conductive object (like a finger). I can not make out the meaning of panel 6i, is there an object on the touch panel? why are all the pixel activated (relative change in capacity is non-zero everywhere).

In summary, the authors improved their manuscript, but there are still very many inaccuracies, incomplete information, and a somewhat rushed presentation of the demos that need attention.

The authors may as well carefully check the whole manuscript on language, as there are a plethora of spelling and grammar mistakes, and even missing words. "frack energy" in SI figure 16 i.e., and many more.

In the intro, "we have demonstrated its flexible ionotronic devices for an ionic conductive substrate, an impendent and a capacitive touch sensor." what does the word "impendent" mean in that context?

I thus recommend the authors carefully address the remaining issues and fill in the remaining gaps in details on their experiments.

Response to Reviewers' Comments

Reviewer #1 (Remarks to the Author):

I would like to support the acceptance of this manuscript.

Response:

We appreciate the reviewer's positive comment on our work very much.

Reviewer #2 (Remarks to the Author):

All concerns raised by the reviewers have been well addressed and thus the manuscript is now at a great state. Please consider to publish as is.

Response:

We are very grateful for the reviewer's recommendation and support.

Reviewer #3 (Remarks to the Author):

In their revision, the authors have addressed several comments of the reviewers. Some are clear and nicely answered, for example the information on cyclic mechanical load is now relatively complete. Some other experiments need however further clarification.

Response:

We sincerely appreciate the reviewer's positive comments and valuable suggestions. Other experiments on the fracture energy tests and the presentation of demos have further clarified in the comments below and the revised manuscript. We hope that you will find our responses sufficient and satisfying.

For once, i could not completely deduce from the author's reply and the revised manuscript, how in detail the fracture toughness tests were performed and how the values for Gamma were calculated. In particular, it is not clear to me how the crack propagation length $\lambda(c)$ or L_c was determined. Typically, the notched sample is stretched until the pre-formed crack propagates, and this is then taken as L_c ($\lambda(c)$) and serves as upper limit for the integrated area in the stress-strain curve of the un-notched sample. (Like in the works of Suo et al.).

Response:

Thanks to the reviewer for the valuable and helpful guidance. For fracture energy tests, pure shear tests were performed with 1kN load cell at a stretching rate of 100 mm min^{-1} (*J. Polym. Sci.* **1953**, 10, 291-318, *J. Appl. Mech.* **2020**, 87, 031002, *Nat. Commun.* **2022**, 13, 2279). This method required two sets of samples with the same size of the width of 50 mm, the thickness of 0.7 ± 0.2 mm, and the gauge height of 10 mm. One set of samples were precut with a crack of 20 mm width, the other set was without crack. (*Extreme Mech. Lett.* **2021**, 48, 101434; *J. Mech. Phys. Solids* **2020**, 134, 103751; *Eng. Fract. Mech.* **2018**, 187, 74-93; *J. Appl. Mech.* **2020**, 87, 031002). The initial height of each sample is denoted as H , and the height in the deformed state changes to λH , where λ is the vertical stretch applied to the sample.

As you suggested, the critical stretch (λ_c) for the notched sample was obtained from when the crack started to propagate. We studied the calculation of the crack propagation

length $\lambda(c)$, as shown in Fig. R1 (*Eng. Fract. Mech.* **2018**, 187, 74-93) and R2 (*J. Appl. Mech.* **2020**, 87, 031002) in the works of Suo. The corresponding unnotched samples were stretched until λc . The fracture energy value (Γ) is obtained by multiplying the elastic strain energy density $W(\lambda c)$ with the initial height H , $\Gamma = HW(\lambda c)$, where $W(\lambda c)$ is obtained by integrating the area under the stress-strain curve of the unnotched sample, H is gauge length.

The cut sample with the same geometry was prepared. A 20-mm crack was cut at the edge by using a razor blade. We applied the same load as that for the uncut samples and observed the opening and propagation of the crack (Fig. 6c). The critical stretches λ_c when the cracks start to propagate are 2.35, 2.25, and 2.07 for samples prepared in solutions of different AAM concentrations. For both cut and uncut samples, 3 measurements were repeated for each AAM concentration and the average results are plotted in Fig. 6g-i. The deviation is within 10%.

For the pure shear test, the stress-stretch relation with the neo-Hookean model is given by

$$s = \mu(\lambda - \lambda^{-3}), \quad (3)$$

Fitting the beginning portion of the curves in Fig. 6d-f gives the shear modulus of all the hydrogels roughly 100 kPa (see Appendix A for fitting).

For the pure shear test, the energy release rate takes the form [79].

$$G = HW(\lambda) \quad (4)$$

where H is the distance between the two grippers when the sample is undeformed, $W(\lambda)$ is the energy per volume of the uncut samples, and λ is the vertical stretch. The energy density $W(\lambda)$ is obtained by integrating the area below the stress-stretch curves of the uncut samples (Fig. 7a).

Figure R1. *Eng. Fract. Mech.* **2018**, 187, 74-93

the deformed state changes to λH , where λ is the vertical stretch applied to the sample. The horizontal displacement is fixed. The stress-stretch curves of the uncut and cut samples with five different chain lengths are plotted in Fig. 4. For the cut samples, as the applied stretch increased, the crack gradually opened up and then started to advance at a critical stretch λ_c .

The energy release rate G for the pure shear configuration is

$$G = HW(\lambda) \quad (2)$$

where $W(\lambda)$ is the free energy density of the uncut samples, calculated from the integrated area below the stress-stretch curve. The fracture toughness λ is

$$\Gamma = HW(\lambda_c) \quad (3)$$

Figure R2. *J. Appl. Mech.* **2020**, 87, 031002

Now, to my understanding, from the text of the authors, it appears the notched samples were stretched until they break, and this was taken as $\lambda(c)$. Such an approach would significantly over-estimate the fracture toughness of the material, and the reported values that exceed 200 kJ/m² seem much too large even for Polyurethanes. An exact description of the measurement procedure is required and would be much appreciated. Please clearly define all variables and formulas used. Including information on standard deviation and number of samples measured.

Response:

Thanks to the reviewer for the critical comments. In our experiments, when the crack propagation began, the notched sample underwent fast fracture (Movie S1), so the stretch λ at the break is close to that of the crack propagation, which was taken as λ_c , as reported in the literatures of Suo (*Extreme Mech. Lett.* **2017**, 10, 24-31) and Wang (*Nat. Commun.* **2022**, 13, 2279).

by stretching a sample containing a pre-cut crack under monotonic load, at a speed of 30 mm/min (Fig. 3(c)). The experiment measured the critical stretch λ_c , at which the pre-cut crack turned into a running crack. That is, at the critical stretch, λ_c , the sample underwent fast fracture, and the energy release rate G reached the fracture energy Γ . The expression

$$HW(\lambda_c) = \Gamma \quad (2)$$

relates the fracture energy to the measured quantities. Our

Figure R3. *Extreme Mech. Lett.* **2017**, 10, 24-31

For fracture energy tests, pure shear tests were performed at a stretching rate of 10 mm min^{-1} , the quadrate specimens are 30 mm wide, 2 mm thick, with a gauge length of 4.5 mm and a crack width of 5 mm. The critical stretch (ϵ_c) for the notched specimens was obtained from the stretch at the break. The pairing unnotched specimens were stretched until $\lambda = \epsilon_c$. The fracture energy value (F) was obtained by multiplying the area under the stress-strain curve of the unnotched specimens with the initial clamp distance, $F = HW(\epsilon_c)$, $W(\epsilon_c)$ is the elastic strain energy density of the unnotched specimen when $\lambda = \epsilon_c$, H is gauge length.

Figure R4. *Nat. Commun.* **2022**, 13, 2279

Therefore, this approach is reasonable to estimate the fracture energy of the material. Furthermore, due to the diversity of its structure, the fracture energy of the increasing Polyurethane systems has been reported to approach or even exceed 200 kJ m^{-2} , as shown in the literatures (*Adv. Mater.* **2020**, 2005759; *Mater. Horiz.* **2021**, 8, 2742-2749; *Adv. Mater.* **2021**, 2101498; *Macromolecules* **2022**, Accepted).

Serving as rigid nanofillers capable of deformation and disintegration under an external force, the dynamic hierarchical domains can strengthen the elastomers and significantly enhance their toughness and fracture energy. As a result, the elastomers exhibit a tensile strength of $\approx 52.4 \text{ MPa}$, a toughness of $\approx 363.8 \text{ MJ m}^{-3}$, and an exceptional fracture energy of $\approx 192.9 \text{ kJ m}^{-2}$. Further-

Figure R5. *Adv. Mater.* **2020**, 2005759

H-bonds and ionic bonds in the network. The WPU elastomer demonstrated that the microphase separation structure contributes to an ultrahigh tensile strength (≈ 58 MPa), super toughness (≈ 456 MJ m⁻³), and **unprecedented record fracture energy (≈ 320 kJ m⁻²)**. Due to the dynamic reconstruction of reversible

Figure R6. *Mater. Horiz.* **2021**, 8, 2742-2749

elastomer has the highest tensile strength (ultimate engineering stress, 75.6 MPa) ever recorded for polymeric elastomers, rendering it the strongest and toughest healable elastomer thus far. The hyper-robust elastomer exhibits superb crack tolerance with **unprecedentedly high fracture energy (215.2 kJ m⁻²)** that even exceeds that of metals and alloys, and superhigh

Figure R7. *Adv. Mater.* **2021**, 2101498

poly(urethane-urea) elastomers with high mechanical strength, stretchability, elasticity, and excellent damage resistance, damage tolerance, and healability are fabricated by cross-linking polycaprolactone (PCL) chains with hydrogen-bond arrays. The elastomer, which is denoted as PU-ASC, has a tensile strength of ~ 72.6 MPa, recovery strain of $\sim 500\%$, and **fracture energy of ~ 161 kJ m⁻²**. Moreover,

Figure R8. *Macromolecules* **2022**, Accepted

As suggested, we have repeated for three times on different batches of samples, as shown in Fig. R9. The average value of fracture energy is 195.99 kJ m⁻², and the standard deviation is 5.09%. Fig. R9(I) as the data closest to the average value is taken as Fig. 3g. Besides, an exact description of the measurement procedure was added in the revised manuscript as following.

Figure R9. Stress-strain curves of three sets of intact DSE-3 sample and one with the precut crack

“Fig. 3g showed the stress-strain curves of the unnotched and notched samples. The fracture energy⁶¹ of the DSE-3 sample was calculated to be as large as 201.29 kJ m⁻² by the formula of $\Gamma = HW(\lambda_c)$, where λ_c is the critical stretch of the crack propagation, $W(\lambda_c)$ is the elastic strain energy density, which is obtained by integrating the area under the stress-strain curve of the unnotched sample at the λ_c , H is gauge length.”

(Line 207-211)

“For fracture energy tests, pure shear tests were performed with 1 kN load cell. This method required two sets of samples with the same size of the width of 50 mm, the thickness of 0.7 ± 0.2 mm, and the gauge height of 10 mm. One set of samples were precut with a crack of 20 mm width, the other set was without crack. The fracture energy value (Γ) is calculated by the formula of $\Gamma = HW(\lambda_c)$, where λ_c is the critical stretch of the crack propagation, $W(\lambda_c)$ is obtained by integrating the area under the stress-strain curve of the unnotched sample at the λ_c , H is gauge length. The results were repeated for three times on different batches of samples” (Line 435-441 in the Methods section)

As for the demos, i am sorry, but i still don't fully understand how the heart shaped LED circuit works. Now, the authors state that the LEDs are "powered by an 220V AC electric field), there is however still no information on how the LEDs are contacted. Are they simply placed on top of a strip of ionic conductor? in this case, does the required voltage difference for driving the LEDs simply stem from the difference in resistance between the legs of the LED? That seems a highly inefficient and unpractical approach to power LEDs, even with ionic conductors. What then is the driving frequency? There is still not much detailed information of the above kind in the methods section.

Response:

We apologize for not clearly expressing the principle of how the heart shaped LED circuit works. As suggested, the poles of the chip LED were embedded into the DSICE ionic conductor, and the chip LED keep a parallel relationship with the ionic conductor. The prepared “heart-shaped” pattern of 18 chip LEDs formed a closed circuit with the DSICE ionic conductor under an AC electric field at a fixed frequency. In addition to the resistance of LEDs, the required voltage difference for driving the LEDs mainly derived from the bulk resistance of DSICE. For the parallel relationship between the chip LED and DSICE ionic conductor, DSICE can carry more LEDs because current flows through LEDs more easily in the parallel circuit. The more LEDs, the more efficient it is. Moreover, DSICE features flexible, stretchable and self-healing, thus

preparing flexible display devices through the chip LEDs designing various pattern. On the other hand, the excellent self-healing capacity of DSICE is beneficial to improve the reliability and prolong the service life of the material. Therefore, DSICE as a novel ionic conductor has potential practical value for the flexible display devices application.

As suggested, the driving frequency is 50 Hz. Besides, we updated the more detailed information in the methods section. More details were added in the revised version as following.

“The poles of the chip LEDs were embedded into the DSICE ionic conductor, and the chip LED keep a parallel relationship with the ionic conductor. The prepared “heart-shaped” pattern of 18 chip LEDs formed a closed circuit with the DSICE under an AC 220 V electric field at the fixed frequency of 50 Hz.” (Line 318-321)

In addition, though the touch panel demos have been improved, i find it very difficult to make out the readings on the photographed impedance analyzers in the small pictures. For example, figure 6c does not seem to have much value other than showing that the impedance analyzer is switched on.

Response:

We appreciate the reviewer for the positive comment and apologize for the small and unclear photographs. Fig. 6c was the plots of the absolute impedance ($|Z|$) as a function of the frequency (f) of DSICE-based touch sensor at four different states and did not show any photograph. Therefore, we assume you were referring to Fig. 6j. As for the phtographs in the Fig. 6j, we have corrected it to Fig. 6h (I) (II) (III), which presented the capacitance value of the touch position where a finger is before and after touching.

Fig. 6. (h) The actual test photographs and mapping showing the localized change in capacitance due to a touch by a finger.

The same goes for figure S22, here one can also barely make out the readings on the photographed screens of the impedance analyzer. Some of this data is plotted in figure 6, right? How would one discriminate touch positions from the impedance spectra? it seems position detection with this kind of sensor seems challenging.

Response:

Thanks for the reviewer's careful guidance and helpful suggestion. We agree with the reviewer's confirmation of the relationship between Fig. 6 and Fig. S22. Some data of Fig. S22 was plotted in Fig. 6c to Fig. 6e, thus making Fig. S22 valueless. In addition, the test monitor and the samples were not in the same dimension, resulting in the readings that were too small to make out. Therefore, Fig. S22 was removed.

For the impedance-based touch sensor, we would like to demonstrate the signals of DSICE in different states (original, touched, stretched, stretched and touched) can be identified by impedance spectra, which cannot realize the position detection. Therefore, we agree with the reviewer's viewpoint. In order to realize the position detection of touch sensing, we demonstrated the capacitive touch sensor below.

“Furthermore, we presented a capacitive touch sensor to realize the position detection of touch sensing.” (Line 351)

If the matrix-based touch demo consists of cross-bars of ionic conductors, separated by

a deformable dielectric, then pressing them should also change capacity, not only touching with a conductive object (like a finger). I can not make out the meaning of panel 6i, is there an object on the touch panel? why are all the pixel activated (relative change in capacity is non-zero everywhere).

Response:

We appreciate the reviewer's constructive comments. As the reviewer thought, for the capacitive touch demo with cross-bars separated by a deformable dielectric, pressing them would lead to the increased capacitance between the electrodes, which is mainly attributed to the decreased thickness and the increased area of the deformable dielectric layer. In addition, the touch object doesn't have to be conductors, and non-conductors can also be (*Adv. Mater.* **2014**, 26, 7608–7614).

When external forces deform the dielectric, the capacitance of this part of the circuit increases (Figure 1d,e). A measurement of this change in capacitance enables the ionic skin to

Figure R10. *Adv. Mater.* **2014**, 26, 7608–7614

But for the capacitive touch sensor in the manuscript, we aim to detect the position of a finger in a touch panel. The proximity and light touch of a finger causes a drop in the capacitance value between the electrodes. The drop capacitance occurs because the finger acts as the third electrode, which capacitively couples to one electrode and reducing the shared charge between the two electrodes (*Sci. Adv.* **2017**, 3, e1602200). Besides, the touch object needs to be conductors, and the finger is an ionic conductor.

In this implementation, each array element is composed of a disc-shaped electrode and its interconnections (Fig. 1B, red) separated from a loop electrode (Fig. 1B, blue) by a dielectric layer. The loop and disc coupling allows for better vertical projection of the field than a simple crossing of lines. The finger acts as a third electrode, which capacitively couples to the sensor element, as represented by the variable capacitor C_F in Fig. 1B, reducing the coupling between electrodes, C_M . These loop and disk elements are placed in a sensor array (depicted in Fig. 1C) made of transparent and deformable materials, as evident in Fig. 1 (D)

Figure R11. *Sci. Adv.* **2017**, 3, e1602200

We are sorry for the unclear expression of Figure 6i. Nothing on the touch panel in Figure 6i, which is the observed impedance change mapping of a finger touching position 1 (Row 2, Col 2). Since the LCR meter can only connect two wires, one crossover point, we simplified the connecting wire counts and only connected Row 2, Col 2, forming the cross position 1. The 16 intersections were touched by a finger in turn, finding that the maximum capacitance change happens at the intersection of Row 2 and Col 2 that is the cross position 1, and the capacitance changes were relatively small at other intersections. The largest capacitance change corresponds to the position of a finger touching on the panel. All the pixel is activated because they suffer from the field disturbance in the position mapping.

Given the above simplified test method is not easy to understand, we have modified the test method to detect the position of the finger on the touch panel. The 4×4 cross-bars array can be made up to 8 lines with 4 on each axis, generating 16 pixels. We detected each combination of row and column electrodes sequentially when a finger fixed on a pixel, and the capacitance changes of all pixels were determined to create a mapping, and the capacitance changes of all pixels were determined to create a mapping. In this case, the maximum capacitance change in the mapping means the position where

a finger touches. This approach is similar to mutual capacitance sensing of which all connecting wires is integrated into one circuit for program detection. Since our LCR meter can only detect one pixel at a time, we adopt the method of detecting each pixel in turn and create a mapping of the capacitance change. Therefore, to demonstrate the work principle of the capacitive touch sensor more clearly, we reworked the experiment of detecting the position of a finger, as shown in Fig 6h. And detailed information was depicted in the revised version.

“In this implementation, the 4×4 cross-bars array can be made up to 8 lines with 4 on each axis, generating 16 pixels. A finger touched a fixed pixel, the capacitance of each combination of row and column electrodes, which was connected with the LCR meter, was detected sequentially. The resulting capacitance changes of all pixels were determined to create a mapping. In this case, the maximum capacitance change value in the mapping means the position where a finger touches. Fig. 6h showed the mapping of a finger touching different positions. It can be observed that the capacitance changes at the intersection of Row 1 and Col 1, that is pixel (1,1), Row 2 and Col 2 (2,2), and Row 3 and Col 3 (3,3) are maximum from Fig. 6h (i) (ii) (iii), and the capacitance decreases by 45.9%, 50.1%, 60.9% from Fig. 6h (I) (II) (III), indicating a finger touched pixel (1,1), (2,2) and (3,3) separately. The neighboring pixels exhibited a certain sensitivity due to the fringe field of the finger.” (Line 358-367)

Fig. 6. (g) The work principle and schematic of the DSICE capacitive touch sensor with a 4×4 cross-bars array. The finger acts as the third electrode, which capacitively couples to one electrode (blue), as represented by the variable capacitor C_F , thus reducing the coupling between two electrodes C_M . (h) The actual test photographs and mapping showing the localized change in capacitance due to a touch by a finger.

Instructions:

All the pixel is activated because they suffer from the field disturbance in the position mapping. Besides, the response of the other pixels in the same column is relatively sensitive because the column that a finger touched was connected with the field. The large fringe field of the finger brings about a slight large change in capacitance. But it has no effect on the maximum capacitance change which is used to detect the position where a finger is.

In summary, the authors improved their manuscript, but there are still very many inaccuracies, incomplete information, and a somewhat rushed presentation of the demos that need attention.

Response:

We appreciate the reviewer's positive comment and apologize for many inaccuracies, incomplete information, and a somewhat rushed presentation of the demos. In response to the above problems, we have added accurate and complete information, and revised the presentation of demos carefully in the revised manuscript.

The authors may as well carefully check the whole manuscript on language, as there are a plethora of spelling and grammar mistakes, and even missing words. "frack energy" in SI figure 16 i.e., and many more.

Response:

We are so sorry for the language mistakes in the whole manuscript. We have revised the mistakes in Fig. S16. Also, we have gone through and revised the entire manuscript to avoid similar mistakes of spelling and grammar mistakes, and even missing words carefully. All the changes were highlighted in yellow in the revised version.

In the intro, "we have demonstrated its flexible ionotronic devices for an ionic conductive substrate, an impendent and a capacitive touch sensor." what does the word "impendent" mean in that context?

Response:

We apologize for the misspelling of the word "impedance". In this sentence, we

would like to express the meaning of “an impedance-based touch sensor”. To avoid spelling and grammar mistakes, we have revised the misspelling of “impence” in Line 78.

“we have demonstrated its flexible ionotronic devices for an ionic conductive substrate, **an impedance-based** and a capacitive touch sensor.” (Line 78)

I thus recommend the authors carefully address the remaining issues and fill in the remaining gaps in details on their experiments.

Response:

We are deeply grateful for the reviewer’s the careful guidance and valuable suggestions, which is very helpful to improve the quality of our manuscript. We have carefully addressed the remaining issues and added the relative details on the experiments.

Reviewers' Comments:

Reviewer #3:

Remarks to the Author:

This time, the authors have really taken my comments to heart and provided thorough answers. The final revision seems correct now, and adds essential new information.

I am especially happy about the more complete characterization of the material's toughness, including the necessary information to interpret the data. The application demos have also improved in this revision, as has the quality of the manuscript all together.

I would now recommend publication.

Response to Reviewers' Comments

Reviewer #3 (Remarks to the Author):

This time, the authors have really taken my comments to heart and provided thorough answers. The final revision seems correct now, and adds essential new information.

I am especially happy about the more complete characterization of the material's toughness, including the necessary information to interpret the data. The application demos have also improved in this revision, as has the quality of the manuscript all together.

I would now recommend publication.

Response:

We sincerely appreciate the reviewer's careful guidance and valuable help for improving our manuscript. Moreover, we thank the reviewer for your recommendation and approval again.